# TriSpec: Ternary Speculative Decoding via Lightweight Proxy Verification

## Abstract

The enhanced reasoning capabilities of Large Language Models (LLMs) have led to longer response sequences, yet the inference efficiency is fundamentally limited by their serial, autoregressive generation. Speculative decoding (SD) offers a powerful solution, providing significant speed-ups through its lightweight drafting and parallel verification mechanism. While existing work has made considerable progress in improving the draft model's generation efficiency and alignment, this paper boosts SD from a new angle: the verification cost. We propose TriSpec, a novel ternary SD framework with proxy verification. At its core, TriSpec introduces a lightweight proxy model that handles the initial verification. This proxy significantly reduces computational cost by approving easily verifiable draft sequences and only engages the full target model when encountering uncertain tokens, thus striking an optimal balance between efficiency and quality. TriSpec can be integrated with state-of-the-art SD methods like EAGLE-3 to further reduce verification costs, achieving greater acceleration. Extensive experiments on the Qwen3 and DeepSeek-R1-Distill-Qwen/LLaMA families show that TriSpec achieves up to 30% speedup over standard SD, with up to 50% fewer target model invocations while maintaining comparable accuracy.

## 1 Introduction

Large language models (LLMs) (Yang et al., 2025; Liu et al., 2024; Grattafiori et al., 2024) have made remarkable success on complex tasks, with recent advances in reasoning leading to longer and more sophisticated responses. This progress, however, is often hampered by the inherent latency of their serial, autoregressive generation process. Speculative decoding (SD) (Leviathan et al., 2023) addresses this efficiency bottleneck with a draft-then-verify paradigm. A small, fast drafter proposes candidate tokens, which the large target model then verifies them in a single parallel pass, significantly accelerating inference without compromising output quality.

Research on speculative decoding has predominantly focused on optimizing the drafting stage, following two main avenues: (i) reducing the drafter's size to accelerate generation, and (ii) enhancing draft quality to improve acceptance rates. Medusa (Cai et al., 2024) and EAGLE (Li et al., 2024a) adopt lightweight drafting heads and a single lightweight draft layer, respectively, shortening the compute path and markedly increasing drafting throughput. Building on this design, subsequent work further raises acceptance rates by more judiciously allocating draft positions and lengths (Li et al., 2024b; Zhang et al., 2024b) and by exploring improved training paradigms for the drafter (Zhang et al., 2024a; Li et al., 2025; Liu et al., 2024). In addition, several verification-side approaches (Bachmann et al., 2025; Narasimhan et al., 2024) relax verification constraints to likewise improve acceptance rates, which in essence is still aimed at identifying high-quality drafts.

An analysis of the end-to-end latency (Lemma 1) in the speculative decoding paradigm reveals a critical oversight: while existing work has effectively reduced latency with more lightweight and better-aligned drafters, the *verification cost* has been largely ignored. This factor has now emerged as a critical bottleneck limiting further efficiency gains. This observation inspired our investigation into a more lightweight verification process, exploring *whether the verification role of the large target model could be effectively handled by a much more efficient, lightweight verifier*.

Our answer to this question is a qualified "yes". We find that smaller models from the same family as the target model (*e.g.*, Qwen3-1.7B and Qwen3-32B) show significant potential, as they exhibit

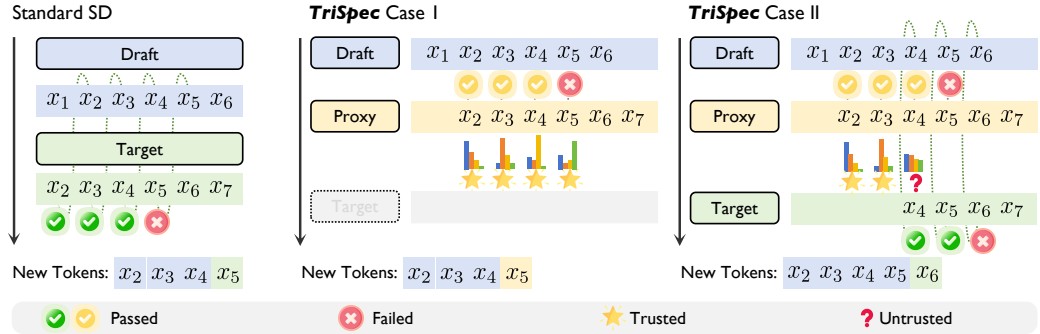

Figure 1: Comparison of standard speculative decoding and TriSpec. Case I: The proxy verifier provides local correction of drafts, bypassing the large-scale target model to gain speed. Case II: If the proxy's verification is deemed untrusted before the first rejection, the corresponding tokens are escalated to the target model for authoritative validation to ensure accuracy. TriSpec employs a margin-based criterion to classify proxy verification as trusted or untrusted.

a high degree of alignment with the output distribution. Consequently, a small model can reliably replaces the target model for verifying the majority of tokens, and the challenge lies in a small subset of tokens where their capabilities diverge. To delineate this capability boundary, we design a specific margin metric that effectively distinguishes between when the smaller model is competent to handle the verification task and when it must defer to the larger target model (Figure 2(b)).

Based on these insights, we propose TriSpec, a ternary speculative decoding paradigm that co-ordinates three models with complementary roles. A single-layer drafter (draft model) provides high-throughput drafting; a lightweight proxy verifier (proxy model) quickly pre-verifies the draft and locally corrects a subset of tokens; and a large verifier (target model) supplies authoritative verification when needed (Figure 1). Extensive experiments on Qwen3 and DeepSeek-R1-Distill-Qwen/LLaMA families demonstrate that TriSpec delivers significant speedups over SOTA speculative decoding methods. In summary, our contributions are summarized as follows:

- We identify a new perspective on SD acceleration, the verification cost. We demonstrate that smaller models from the same family exhibit strong alignment with their larger counterparts, establishing their viability as lightweight and effective proxy verifiers.

- We propose TriSpec, a ternary speculative decoding paradigm that coordinates a drafter, a proxy verifier, and the target model. We design a margin-based routing rule to dynamically delegate verification tasks, engaging the computationally expensive target model only when necessary to ensure accuracy while minimizing overhead.

- Extensive experiments show that TriSpec achieves up to a 30% speedup over standard speculative decoding and reduces invocations of the target model by more than half. This efficiency is achieved with a minimal average accuracy drop of less than 1%.

## 2 BACKGROUND AND ANALYSIS ON SPECULATIVE DECODING

### 2.1 SPECULATIVE DECODING

Standard speculative decoding follows a draft-then-verify paradigm with two models: a draft model $\mathcal{M}_d$ and a target model $\mathcal{M}_t$. Let $\mathcal{V}$ denote a finite vocabulary with size $V := |\mathcal{V}|$, and let the current decoding prefix be $X \in \mathcal{V}^l$, including both the prompt and previously generated tokens.

**Drafting phase.** During drafting, $\mathcal{M}_d$ autoregressively produces $k$ probability distributions and the corresponding sampled tokens:

$$\left(x_1^{(d)}, \mathbf{p}_1^{(d)}\right), \; \left(x_2^{(d)}, \mathbf{p}_2^{(d)}\right), \; \ldots, \; \left(x_k^{(d)}, \mathbf{p}_k^{(d)}\right) \; \leftarrow \; \mathcal{M}_d(X), \tag{1}$$

where each $\mathbf{p}_i^{(d)} \in \mathbb{R}^V$ is the softmax probability distribution produced by the draft model at position $i$ and $x_i^{(d)} \in \mathcal{V}$ is sampled from $\mathbf{p}_i^{(d)}$. For a probability distribution $\mathbf{p}$ and a token $x \in \mathcal{V}$, we write $\mathbf{p}(x)$ for the probability assigned to $x$.

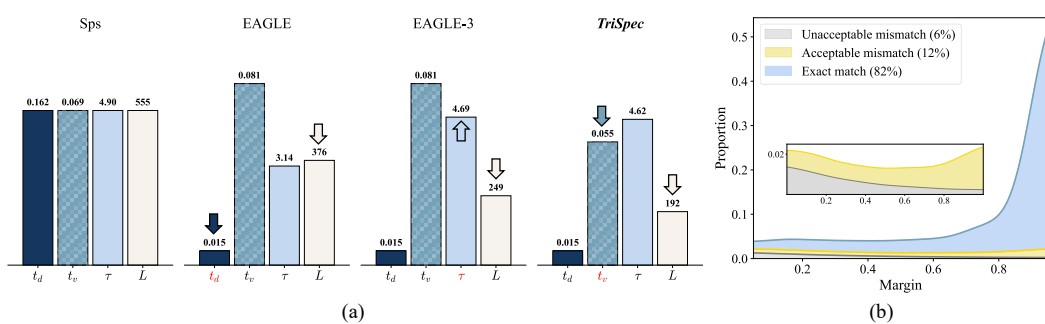

(a)                                              (b)

Figure 2: (a) Latency decomposition of speculative decoding across representative methods. EA-GLE reduces drafting cost $t_d$ via a single-layer design. EAGLE-3 and related methods further enlarge the acceptance length $\tau$ by improving draft token quality. Our proposed TriSpec introduces a new perspective by reducing verification cost $t_v$, thereby lowering overall latency $L$. (b) Distribution of token outcomes versus the proxy's top-1–top-2 probability margin on ShareGPT. Areas are stacked and globally normalized, so the total across all classes sums to 1.

**Verification phase.** In the verification stage, the target model $\mathcal{M}_t$ evaluates drafted tokens through a single parallel forward pass:

$$(x_1^{(t)}, \mathbf{p}_1^{(t)}),\ (x_2^{(t)}, \mathbf{p}_2^{(t)}),\ \ldots,\ (x_{k+1}^{(t)}, \mathbf{p}_{k+1}^{(t)})\ \leftarrow\ \mathcal{M}_t(x_1^{(d)}, x_2^{(d)}, \ldots, x_k^{(d)} \mid X), \tag{2}$$

where $x_i^{(t)} \in \mathcal{V}$ and $\mathbf{p}_i^{(t)} \in \mathbb{R}^V$ denote the token and the probability distributions generated by the target model, respectively. The verification process proceeds sequentially from left to right: each token $x_i^{(d)}$ is accepted with probability $\min(1,\, \mathbf{p}_i^{(t)}(x_i)/\mathbf{p}_i^{(d)}(x_i))$ and appended to the output if accepted. Upon the first rejection, all remaining drafted tokens are discarded, and a replacement is sampled from the normalized positive residual of the target distribution $\mathrm{norm}(\max(0,, \mathbf{p}_i^{(t)} - \mathbf{p}_i^{(d)}))$. After resampling, a new round of speculative decoding starts from the updated prefix. Prior work (Chen et al., 2023) has rigorously established that speculative sampling reproduces the exact output distribution of naive target-model decoding and is therefore provably lossless.

## 2.2 Latency Analysis of Speculative Decoding

**Lemma 1** (Latency decomposition for SD). *Consider a speculative decoding process that generates $N$ new tokens. Let $\tau$ denote the average acceptance length (tokens accepted per verification round), and let $t_d$, $t_v$, and $t_o$ be the average per-round drafting time, verification time, and other overhead, respectively. Then the end-to-end latency satisfies*

$$L\ =\ \frac{N}{\tau}\big(t_d + t_v + t_o\big). \tag{3}$$

In practice, $t_o$ is typically small and approximately constant compared to $t_d$ and $t_v$, and since standard SD is theoretically lossless (Chen et al., 2023), $N$ can be considered constant. Consequently,

$$L\ \propto\ \frac{t_d + t_v}{\tau}. \tag{4}$$

**Remark 1.** Recent progress on latency reduction has primarily been achieved by decreasing $t_d$ and increasing $\tau$. Early SD employed a separate lightweight model as the drafter (Leviathan et al., 2023), but the use of multi-layer attention rendered the drafting stage latency-dominant (Figure 2(a), Sps). Lightweight drafter designs, such as the drafting heads in Medusa (Cai et al., 2024) and the single drafting layer in EAGLE (Li et al., 2024a), significantly reduce $t_d$ while maintaining a high $\tau$ (Figure 2(a), EAGLE), making them the prevailing approach. To further increase $\tau$, subsequent work has improved draft-token prediction accuracy by exploring more effective draft-tree architectures (Li et al., 2024b) and training schemes (Zhang et al., 2024a; Li et al., 2025) (Figure 2(a), EAGLE-3). With Multi-Token Prediction (MTP) (Liu et al., 2024) integrating draft-model training into the pre-training stage, the acceptance rate of draft tokens can even reach as high as 90%.

**Remark 2.** Little progress has been made in directly reducing the per-round verification time $t_v$. In state-of-the-art pipelines (e.g., EAGLE-3), verification has become the dominant source of per-round

latency, as each target-model invocation traverses a long computational path with a large parameter budget. A straightforward idea is to employ a lightweight verifier to offload part of the verification from the target, thereby reducing the average verification time across rounds.

## 3 TRISPEC

In this section, we propose TriSpec, a ternary speculative decoding framework that integrates a single-layer drafter (draft model), a lightweight proxy verifier (proxy), and the target model.

### 3.1 PROXY CHOICE: SAME-FAMILY SMALLER MODELS

We show that smaller models from the same family are strong candidates for lightweight proxy verifiers capable of reducing verification costs, based on the following two key properties.

**Strong Alignment.** Because SD verifies tokens individually, we assess token-level alignment between the proxy and the target model on the ShareGPT dataset using Qwen3-family models. At each position, we compare the proxy's prediction of the next token with the target's and classify the results into three categories: exact match (same token), acceptable mismatch (different token but considered acceptable by a stronger LLM), and unacceptable mismatch (otherwise). The experiment details are provided in Appendix C.2. Under this protocol, the proxy achieves 82% exact match with the target (EAGLE-3 drafter: 68%), with only 6% of tokens deemed unacceptable, demonstrating a strong alignment at the token level.

**Trustworthy outputs.** Beyond strong alignment with the target, the proxy's outputs are themselves trustworthy. In the standalone example (Appendix C.1), despite phrasing differences, the proxy and target reach the same final answer, indicating that modest token-level discrepancies between them are tolerable. Consistently, the end-to-end relaxed-verification experiments (Appendix C.3) show that limited deviations from the target do not materially affect accuracy. For routing, the key question is which positions can be trusted under proxy verification. Intuitively, when the proxy is more confident in its top-1 prediction, it is less likely to diverge from the target, and its influence on the final answer is smaller. We quantify verification confidence as the margin between the proxy's top-1 and top-2 probabilities at each position. As shown in Figure 2(b), this margin cleanly stratifies tokens: when the margin is large (e.g., $> 0.5$), most proxy predictions are acceptable, whereas unacceptable tokens concentrate near zero. Thus, the margin provides an effective, actionable criterion for classifying the proxy's output as trusted or untrusted.

### 3.2 TERNARY SPECULATIVE DECODING

**Drafter selection and adapter training.** We choose the single-layer drafter architecture proposed by EAGLE (Zhang et al., 2024a; Li et al., 2025; 2024a) as our draft model, since it provides state-of-the-art efficiency in drafting. A central principle of EAGLE is that feature-level autoregression is empirically simpler than token-level autoregression. Accordingly, the drafter conditions on both the immediately preceding token and its corresponding features. In particular, for the first token, it must be seeded with features computed by the proxy or target model.

To support proxy-based verification, the drafter in TriSpec must flexibly accept features from either source. As illustrated in Figure 3(a), we extend the original EAGLE architecture with a one-layer MLP adapter that maps proxy features into the drafter's expected feature space. This adapter can be trained under two regimes: (i) joint training from scratch, where the drafter and adapter are optimized together using proxy and target features; or (ii) adapter-only finetuning, where a pretrained EAGLE drafter's weights are fixed and only the adapter is trained. We provide empirical comparisons of these regimes in Section 4.4.

**Proxy pre-Verification.** After the drafter produces a length-$k$ draft as in Eq. 1, we exploit the lower query cost of the proxy verifier to screen the draft before invoking the target. Concretely, given the drafted pairs $(x_1^{(d)}, \mathbf{p}_1^{(d)}), \ldots, (x_k^{(d)}, \mathbf{p}_k^{(d)})$, the proxy $\mathcal{M}_p$ performs a single parallel pass and returns its own token predictions and distributions:

$$\left(x_1^{(p)}, \mathbf{p}_1^{(p)}\right), \ \left(x_2^{(p)}, \mathbf{p}_2^{(p)}\right), \ \ldots, \ \left(x_{k+1}^{(p)}, \mathbf{p}_{k+1}^{(p)}\right) \ \leftarrow \ \mathcal{M}_p\!\left(x_1^{(d)}, x_2^{(d)}, \ldots, x_k^{(d)} \,\middle|\, X\right), \qquad (5)$$

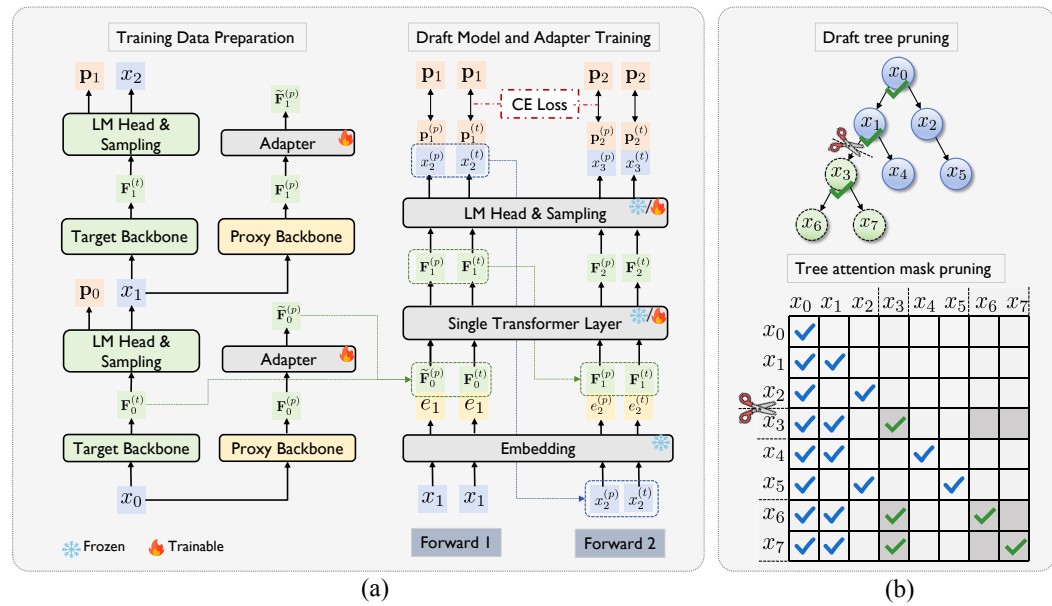

Figure 3: (a) Training pipeline of the draft model and adapter in TriSpec. The green, yellow, and gray modules represent the target, proxy, and drafter components, respectively. (b) Draft tree pruning and tree attention mask mechanisms in TriSpec.

where $x_i^{(p)} \in \mathcal{V}$ and $\mathbf{p}_i^{(p)} \in \mathbb{R}^V$ denote the proxy's next-token prediction and its softmax distribution, respectively. Analogous to standard speculative decoding, the proxy determines token acceptance by comparing its probability distribution with that of the drafter. Formally, for each draft token $x_i^{(d)}$, we define a Bernoulli acceptance variable

$$s_i \sim \text{Bernoulli}\left( \min\left\{ 1, \frac{\mathbf{p}_i^{(p)}(x_i^{(d)})}{\mathbf{p}_i^{(d)}(x_i^{(d)})} \right\} \right), \quad i = 1, \dots, k,$$

which in the greedy case reduces to $s_i = \mathbf{1}\{x_i^{(p)} = x_i^{(d)}\}$. Then, the proxy-side acceptance length is

$$\tau_a = \min\left( \{ i - 1 \mid 1 \leq i \leq k, \ s_i = 0 \} \cup \{k\} \right). \tag{6}$$

**Margin-based routing strategy.** Although informative, proxy-side verification alone is not sufficiently reliable. Empirically (see §3.1), we observe that the margin between the proxy's top-1 and top-2 probabilities correlates strongly with reliability. Motivated by this, we introduce a margin-based binary predicate $g : \mathbb{R}^V \to \{0, 1\}$ to assess whether the proxy's verification can be trusted:

$$g\left(\mathbf{p}_i^{(p)}\right) = \mathbf{1}\left\{ \text{top}_1\left(\mathbf{p}_i^{(p)}\right) - \text{top}_2\left(\mathbf{p}_i^{(p)}\right) \geq \lambda \right\}, \qquad i = 1, \dots, k+1, \tag{7}$$

where $\text{top}_1(x)$ and $\text{top}_2(x)$ denote the largest and second-largest entries of $x$, and $\lambda \in (0, 1)$ is a tunable threshold controlling the accuracy–latency trade-off. To measure how long we can rely on the proxy before escalation, we define the proxy-trusted prefix length as the longest initial segment that passes the margin test:

$$\tau_m = \min\left( \{ i - 1 \mid 1 \leq i \leq k+1, \ g(\mathbf{p}_i^{(p)}) = 0 \} \cup \{k+1\} \right). \tag{8}$$

In particular, if $g(\mathbf{p}_i^{(p)}) = 1$ for all $i$, then $\tau_m = k + 1$, indicating that the proxy's verification is trusted for the entire draft and can reliably produce the $(k+1)$-th token.

Then, the procedure of verification can be divided into two cases, depending on the necessity of invoking the target:

(i) If $\tau_a < \tau_m$, the proxy's verification remains trustworthy at the first rejection point, and it can complete the verification round without invoking the target model (Figure 1 center). Accordingly, we

---

**Algorithm 1** TriSpec in a single round

---

1: **Inputs:** Draft model $\mathcal{M}_d$, Proxy $\mathcal{M}_p$, Target $\mathcal{M}_t$, input prefix $X$, margin-based threshold $\lambda$
2: **for** $i = 1$ to $k$ **do**      ▷ Sample k draft tokens from $\mathcal{M}_d$ autoregressively
3:      $\mathbf{p}_i^{(d)} \leftarrow \mathcal{M}_d(X + [x_1, \ldots, x_{i-1}])$
4:      $x_i \sim \mathbf{p}_i^{(d)}$
5: **end for**
6: $\tau_a,\ [\mathbf{p}_1^{(p)}, \ldots, \mathbf{p}_{k+1}^{(p)}] \leftarrow \text{Verify}(\mathcal{M}_p, X + [x_1, \ldots, x_k])$     ▷ Pre-verification by $\mathcal{M}_p$
7: $g(\mathbf{p}_i^{(p)}) \leftarrow \mathbf{1}\left\{ \text{top}_1(\mathbf{p}_i^{(p)}) - \text{top}_2(\mathbf{p}_i^{(p)}) \geq \lambda \right\},\ i = 1, \ldots, k+1$     ▷ Margin-based rule
8: $\tau_m \leftarrow \min(\{\, i - 1 \mid 1 \leq i \leq k+1,\ g(\mathbf{p}_i^{(p)}) = 0\} \cup \{k+1\})$     ▷ Proxy-trusted prefix length
9: **if** $\tau_a < \tau_m$ **then**   ▷ Without invoking the target
10:      $x_{\text{bonus}} \sim \mathbf{p}_{\tau_a+1}^{(p)}$
11:      **return** $X + [x_1, \ldots, x_{\tau_a}, x_{\text{bonus}}]$
12: **else**   ▷ invoking the target for authoritative verification
13:      $\tau_t,\ [\mathbf{p}_{\tau_m+1}^{(t)}, \ldots, \mathbf{p}_{k+1}^{(t)}] \leftarrow \text{Verify}(\mathcal{M}_t, X + [x_1, \ldots, x_k])$     ▷ Verification by $\mathcal{M}_t$
14:      $x_{\text{bonus}} \sim \mathbf{p}_{\tau_m+\tau_t+1}^{(t)}$
15:      **return** $X + [x_1, \ldots, x_{\tau_m+\tau_t}, x_{\text{bonus}}]$
16: **end if**

---

accept $x_{1:\tau_a}^{(d)}$ and locally correct the next token by replacing it with $x_{\tau_a+1}^{(p)}$. The generated sequence for this round is $Y = \{x_1^{(d)}, \ldots, x_{\tau_a}^{(d)}, x_{\tau_a+1}^{(p)}\}$.

(ii) If $\tau_a \geq \tau_m$, the proxy's verification is deemed untrusted before the first rejection point, so the verification is thus delegated to the target model (Figure 1 right). We first accept $x_{1:\tau_m}^{(d)}$, prune these tokens from the draft tree (Figure 3(b)), and pass the remaining branches to the target for validation:

$$(x_{\tau_m+1}^{(t)}, \mathbf{p}_{\tau_m+1}^{(t)}),\ \ldots,\ (x_{k+1}^{(t)}, \mathbf{p}_{k+1}^{(t)})\ \leftarrow\ \mathcal{M}_t(x_{\tau_m+1}^{(d)}, \ldots, x_k^{(d)} \mid X, x_{1:\tau_m}^{(d)}), \qquad (9)$$

Applying the standard left-to-right acceptance rule to $\mathbf{p}^{(t)}$ yields the target-verified acceptance length $\tau_t$. The generated sequence for this round is $Y = \{x_1^{(d)}, \ldots, x_{\tau_m+\tau_t}^{(d)}, x_{\tau_m+\tau_t+1}^{(t)}\}$.

## 4 EXPERIMENTS

### 4.1 SETUP

**Models and baselines.** We evaluate TriSpec on three model families: Qwen3, DeepSeek-R1-Distill-Qwen (denoted by DSQ), and DeepSeek-R1-Distill-LLaMA (denoted by DSL). Specifically, we use Qwen3-32B, DSQ-32B, and DSL-70B as the *target models*, with Qwen3-1.7B, DSQ-1.5B, and DSL-8B as the *proxy verifiers*. For each target model, we use the corresponding HASS/EAGLE3 weights as the *draft models*. To ensure fairness, all draft models are trained from scratch on the same data using the official HASS and EAGLE3 codebases. As baselines, we include standard speculative decoding and three methods for relaxed verification proposed by SpecCascade (Narasimhan et al., 2024), plus proxy-only and target-only decoding as reference points. For all SD methods, we follow the EAGLE-2 configuration during inference: draft tree depth 6, expansion top-k 10, and a total budget of 60 drafted tokens. The margin threshold $\lambda$ in TriSpec is set to 0.5. The details of baselines and other implementations are shown in Appendix D.1.

**Benchmarks.** We evaluate on challenging reasoning benchmarks: three mathematical reasoning datasets (GSM8K (Cobbe et al., 2021), MATH500 (Lightman et al., 2023), Gaokao-2023-EN) and two code-generation datasets (HumanEval (Chen et al., 2021), MBPP (Austin et al., 2021)). To further evaluate performance across different domains and on extremely difficult benchmarks, we also include SpecBench (Xia et al., 2024), HotpotQA (Yang et al., 2018), Polymath (Wang et al., 2025b), AIME24&25 (Zhang & Math-AI, 2024), and GPQA-Diamond (Rein et al., 2024). To prevent excessive GPU memory usage, we set the maximum generation length to 8192 tokens. Consistent with prior work (Cai et al., 2024; Li et al., 2025) on speculative decoding, we fix the batch size to 1. Given the inference cost of reasoning models, we randomly select 100 samples of each benchmark.

Table 1: Accuracy and speedup of different methods on five reasoning benchmarks, evaluated with a sampling temperature of 0. "SC" denotes the SpecCascade baseline. "HASS/EAGLE3 + TriSpec" refers to a draft model trained under the HASS or EAGLE3 pipeline and employing TriSpec for proxy-based verification.

| Model | Method | GSM8K | | MATH500 | | GK23-en | | HumanEval | | MBPP | | Avg | |
|---|---|---|---|---|---|---|---|---|---|---|---|---|---|
| | | Acc. | Speedup | Acc. | Speedup | Acc. | Speedup | Acc. | Speedup | Acc. | Speedup | Acc. | Speedup |
| | Single Model (Target,32B) | 98% | 1.00x | 85% | 1.00x | 75% | 1.00x | 87% | 1.00x | 77% | 1.00x | 84.4% | 1.00x |
| | Single Model (Proxy,1.7B) | 86% | 2.36x | 81% | 3.01x | 70% | 3.03x | 81% | 2.93x | 51% | 2.92x | 73.8% -10.6% | 2.85x |
| | HASS | 98% | 2.98x | 86% | 2.27x | 74% | 2.30x | 87% | 2.03x | 77% | 2.18x | 84.4% ±0% | 2.35x |
| | HASS + SC[Chow] | 80% | 3.33x | 55% | 2.60x | 51% | 2.64x | 74% | 2.23x | 56% | 2.33x | 63.2% -21.2% | 2.63x |
| | HASS + SC[OPT] | 76% | 3.25x | 38% | 2.66x | 33% | 2.65x | 49% | 2.30x | 20% | 2.34x | 43.2%-41.2% | 2.64x |
| Qwen3 | HASS + SC[Token] | 98% | 3.10x | 86% | 2.34x | 74% | 2.39x | 87% | 2.15x | 74% | 2.28x | 83.8% -0.6% | 2.63x |
| | HASS + TriSpec | 97% | **3.85x** | 86% | **3.18x** | 75% | **3.26x** | 86% | **2.99x** | 74% | **2.80x** | 83.6% -0.8% | **3.22x** |
| | EAGLE3 | 98% | 3.25x | 87% | 2.73x | 74% | 2.77x | 87% | 2.71x | 76% | 2.77x | 84.4% ±0% | 2.85x |
| | EAGLE3 + SC[Chow] | 95% | 3.39x | 70% | 2.91x | 58% | 2.94x | 76% | 2.82x | 67% | 2.85x | 73.2% -11.2% | 2.98x |
| | EAGLE3 + SC[OPT] | 93% | 3.43x | 58% | 3.06x | 42% | 3.01x | 61% | 2.91x | 49% | 2.87x | 60.6%-23.8% | 3.06x |
| | EAGLE3 + SC[Token] | 98% | 3.34x | 86% | 2.90x | 74% | 2.91x | 87% | 2.78x | 76% | 2.85x | 84.2% -0.2% | 2.96x |
| | EAGLE3 + TriSpec | 98% | **4.18x** | 88% | **3.49x** | 74% | **3.55x** | 87% | **3.27x** | 74% | **3.24x** | 84.2% -0.2% | **3.55x** |
| | Single Model (Target,32B) | 93% | 1.00x | 86% | 1.00x | 74% | 1.00x | 85% | 1.00x | 72% | 1.00x | 82.0% | 1.00x |
| | Single Model (Proxy,1.5B) | 72% | 3.09x | 76% | 3.32x | 62% | 3.33x | 54% | 3.22x | 38% | 3.32x | 60.4% -21.6% | 3.26x |
| | HASS | 93% | 3.28x | 88% | 2.99x | 72% | 2.90x | 84% | 2.78x | 73% | 3.02x | 82.0% ±0% | 2.99x |
| | HASS + SC[Chow] | 77% | 3.46x | 51% | 3.22x | 44% | 3.06x | 75% | 2.85x | 62% | 3.10x | 61.8% -20.2% | 3.14x |
| | HASS + SC[OPT] | 77% | 3.59x | 39% | 3.20x | 20% | 3.18x | 41% | 2.88x | 18% | 3.12x | 39.0%-43.0% | 3.19x |
| DSQ | HASS + SC[Token] | 91% | 3.41x | 85% | 3.13x | 69% | 2.96x | 84% | 2.80x | 72% | 3.04x | 80.2% -1.8% | 3.07x |
| | HASS + TriSpec | 91% | **3.76x** | 87% | **3.41x** | 74% | **3.33x** | 85% | **2.98x** | 71% | **3.24x** | 81.6% -0.4% | **3.34x** |
| | EAGLE3 | 93% | 3.42x | 88% | 3.19x | 74% | 3.11x | 86% | 3.09x | 72% | 3.27x | 82.6% +0.6% | 3.22x |
| | EAGLE3 + SC[Chow] | 84% | 3.67x | 69% | 3.47x | 51% | 3.43x | 72% | 3.19x | 64% | 3.33x | 68.0% -14.0% | 3.42x |
| | EAGLE3 + SC[OPT] | 81% | 3.69x | 58% | 3.51x | 45% | 3.46x | 54% | 3.23x | 48% | 3.33x | 57.2%-24.8% | 3.44x |
| | EAGLE3 + SC[Token] | 92% | 3.54x | 82% | 3.36x | 72% | 3.26x | 86% | 3.18x | 70% | 3.27x | 80.4% -1.6% | 3.32x |
| | EAGLE3 + TriSpec | 91% | **4.14x** | 88% | **3.69x** | 75% | **3.63x** | 88% | **3.32x** | 69% | **3.57x** | 82.2% +0.2% | **3.67x** |
| | Single Model (Target,70B) | 88% | 1.00x | 90% | 1.00x | 78% | 1.00x | 87% | 1.00x | 79% | 1.00x | 84.4% | 1.00x |
| | Single Model (Proxy,8B) | 77% | 3.71x | 83% | 3.73x | 73% | 4.01x | 80% | 3.89x | 59% | 4.03x | 74.4% -20.0% | 3.87x |
| DSL | EAGLE3 | 88% | 2.83x | 89% | 2.35x | 76% | 2.31x | 84% | 2.49x | 80% | 2.40x | 83.4% -1.0% | 2.48x |
| | EAGLE3 + SC[Chow] | 82% | 2.90x | 81% | 2.50x | 65% | 2.61x | 74% | 2.58x | 71% | 2.57x | 74.6% -9.8% | 2.63x |
| | EAGLE3 + SC[OPT] | 78% | 3.02x | 66% | 2.84x | 48% | 2.93x | 60% | 2.89x | 51% | 2.91x | 60.6% -23.8% | 2.92x |
| | EAGLE3 + SC[Token] | 87% | 2.92x | 88% | 2.43x | 75% | 2.56x | 85% | 2.69x | 78% | 2.65x | 82.6% -1.8% | 2.65x |
| | EAGLE3 + TriSpec | 86% | **3.74x** | 88% | **3.35x** | 77% | **3.36x** | 87% | **3.07x** | 76% | **3.29x** | 82.8% -1.6% | **3.36x** |

**Metrics.** For task performance, we report pass@1 accuracy on all reasoning tasks, while using an LLM judge score for the general-domain task SpecBench. For efficiency, we report throughput (tokens per second, tps) and compute speedup ratio from throughput relative to naive target-only decoding. To better quantify target invocations, we introduce the target-invocation ratio ($r_t$), defined as the number of target model invocations divided by the number of generated tokens. A smaller $r_t$ indicates reduced reliance on the target and a lighter verification footprint.

## 4.2 EFFECTIVENESS OF TRISPEC

Table 1 reports the performance and efficiency of TriSpec under a sampling temperature of 0. On all benchmark, TriSpec maintains performance on both mathematical and coding tasks, with an average loss no greater than 1%. At the same time, proxy-based verification yields higher throughput than standard SD verification. On Qwen3, TriSpec raises the average speedup of HASS from 2.30× to 3.26× and of EAGLE3 from 2.77× to 3.55×, i.e., roughly a 30% relative improvement; on the R1-Distill-Qwen and R1-Distill-LLaMA family, TriSpec likewise delivers about a 15% and 35% gain over the corresponding standard SD pipelines. By contrast, other lossy approaches based on single-layer drafters (e.g., SpecCascade) struggle to balance accuracy and speedup benefits while delivering speedups, which highlights the superiority of TriSpec. Detailed discussions of these methods are provided in the Appendix C.3.

A detailed breakdown of MATH500 and HumanEval is presented in Table 3. To avoid the impact of repeated generation caused by erroneous reasoning, we compare TriSpec only with standard SD. By introducing a proxy verifier to pre-validate drafts, TriSpec substantially reduces target invocations (e.g., 25% vs. 10.24% for HASS and 21.8% vs. 9.95% for EAGLE3 on MATH500 with Qwen3

Table 2: Accuracy and speedup of different methods on five reasoning benchmarks, evaluated with a sampling temperature of 1.

| | | GSM8K | | MATH500 | | GK23-en | | HumanEval | | MBPP | | Avg | |
|---|---|---|---|---|---|---|---|---|---|---|---|---|---|
| Model | Method | Acc. | Speedup | Acc. | Speedup | Acc. | Speedup | Acc. | Speedup | Acc. | Speedup | Acc. | Speedup |
| Qwen3 | Single Model (Target,32B) | 98% | 1.00x | 85% | 1.00x | 76% | 1.00x | 87% | 1.00x | 76% | 1.00x | 84.4% | 1.00x |
| | Single Model (Proxy,1.7B) | 88% | 2.62x | 80% | 2.84x | 73% | 2.90x | 76% | 2.90x | 61% | 2.90x | 75.6% -8.8% | 2.83x |
| | HASS | 96% | 2.61x | 86% | 2.05x | 75% | 2.05x | 88% | 1.96x | 74% | 1.94x | 83.8% -0.6% | 2.12x |
| | HASS + TriSpec | 95% | **3.01x** | 84% | **2.51x** | 77% | **2.54x** | 89% | **2.31x** | 72% | **2.25x** | 83.4% -1.0% | **2.52x** |
| | EAGLE3 | 98% | 2.78x | 84% | 2.55x | 77% | 2.52x | 86% | 2.58x | 74% | 2.52x | 83.8% -0.6% | 2.59x |
| | EAGLE3 + TriSpec | 96% | **3.33x** | 86% | **3.06x** | 76% | **3.13x** | 88% | **2.94x** | 73% | **2.93x** | 83.8% -0.6% | **3.08x** |
| DSQ | Single Model (Target,32B) | 92% | 1.00x | 88% | 1.00x | 77% | 1.00x | 88% | 1.00x | 76% | 1.00x | 84.2% | 1.00x |
| | Single Model (Proxy,1.5B) | 71% | 2.99x | 78% | 2.91x | 66% | 3.15x | 62% | 3.12x | 47% | 3.11x | 64.8% -19.4% | 3.06x |
| | HASS | 90% | 2.97x | 87% | 2.19x | 76% | 2.33x | 90% | 1.98x | 75% | 2.04x | 83.6% -0.6% | 2.30x |
| | HASS + TriSpec | 89% | **3.27x** | 87% | **2.47x** | 77% | **2.74x** | 87% | **2.12x** | 73% | **2.18x** | 82.6% -1.6% | **2.56x** |
| | EAGLE3 | 92% | 3.14x | 85% | 2.40x | 77% | 2.53x | 89% | 2.23x | 76% | 2.29x | 83.8% -0.4% | 2.52x |
| | EAGLE3 + TriSpec | 88% | **3.40x** | 87% | **2.79x** | 79% | **2.92x** | 88% | **2.36x** | 72% | **2.42x** | 82.8% -1.4% | **2.78x** |
| DSL | Single Model (Target,70B) | 87% | 1.00x | 88% | 1.00x | 75% | 1.00x | 87% | 1.00x | 80% | 1.00x | 83.4% | 1.00x |
| | Single Model (Proxy,8B) | 72% | 3.80x | 81% | 3.92x | 70% | 3.95x | 79% | 3.93x | 60% | 3.96x | 72.4% -11.0% | 3.91x |
| | EAGLE3 | 85% | 2.79x | 88% | 1.98x | 76% | 2.05x | 88% | 2.01x | 81% | 1.96x | 83.2% +0.2% | 2.16x |
| | EAGLE3 + TriSpec | 86% | **3.93x** | 89% | **2.74x** | 78% | **3.06x** | 89% | **2.32x** | 78% | **2.39x** | 84.0% +0.6% | **2.89x** |

Table 3: Comparison of per-sample averages across all methods on MATH500 and HumanEval: Target-invocation ratio ($r_t$), average per-round verification time ($t_v$), average acceptance length ($\tau$), Latency($L$), and Speed.

| | | MATH500 | | | | | HumanEval | | | | |
|---|---|---|---|---|---|---|---|---|---|---|---|
| Model | Method | $r_t$ | $t_v$(s) | $\tau$ | $L$(s) | Speed(Tok/s) | $r_t$ | $t_v$(s) | $\tau$ | $L$(s) | Speed(Tok/s) |
| Qwen3 | Single Model (Target,32B) | 100.00% | / | / | 258.76 | 14.24 | 100.00% | / | / | 234.27 | 14.65 |
| | HASS | 25.01% | 0.094 | 4.21 | 115.04 | 32.45 | 27.14% | 0.103 | 3.93 | 119.32 | 29.80 |
| | HASS + TriSpec | **10.24%** | **0.070** | **4.39** | **83.03** | **45.26** | **11.61%** | **0.077** | **4.41** | **78.74** | **43.38** |
| | EAGLE3 | 21.38% | 0.094 | **4.72** | 97.52 | 38.96 | 22.05% | 0.103 | 4.60 | 86.62 | 39.76 |
| | EAGLE3 + TriSpec | **9.95%** | **0.071** | 4.59 | **74.73** | **49.74** | **11.15%** | **0.080** | **4.76** | **71.83** | **48.00** |
| DSQ | Single Model (Target,32B) | 100.00% | / | / | 156.82 | 16.82 | 100.00% | / | / | 163.68 | 17.35 |
| | HASS | 22.59% | 0.068 | **4.50** | 47.92 | 50.29 | 23.55% | 0.072 | **4.16** | 56.75 | 48.40 |
| | HASS + TriSpec | **11.10%** | **0.054** | 4.31 | **43.25** | **57.30** | **14.20%** | **0.061** | 4.05 | **52.90** | **51.15** |
| | EAGLE3 | 20.67% | 0.068 | **4.79** | 46.13 | 53.77 | 21.45% | 0.072 | 4.54 | 51.80 | 53.66 |
| | EAGLE3 + TriSpec | **10.63%** | **0.055** | 4.78 | **42.06** | **60.03** | **13.25%** | **0.064** | 4.54 | **48.92** | **57.02** |
| DSL | Single Model (Target,70B) | 100.00% | / | / | 256.79 | 8.22 | 100.00% | / | / | 254.18 | 8.37 |
| | EAGLE3 | 28.70% | 0.144 | 3.48 | 108.89 | 19.33 | 26.88% | 0.147 | **3.66** | 107.92 | 20.84 |
| | EAGLE3 + TriSpec | **11.60%** | **0.093** | **3.64** | **78.32** | **27.54** | **15.18%** | **0.114** | 3.59 | **87.51** | **25.70** |

family). Although TriSpec introduces a pre-validating pass, the reduced number of target invocations leads to a significant drop in the average per-round verification time $t_v$. Meanwhile, thanks to the proxy's strong alignment with the target, the average acceptance length $\tau$ remains stable. Consequently, the end-to-end latency decreases relative to standard SD.

We further evaluate TriSpec at a sampling temperature of 1.0 on Table 2. Under non-deterministic decoding, TriSpec continues to deliver notable speedup gains while maintaining competitive accuracy, with an average accuracy drop no greater than 2% and speedup improvements ranging from 10% to 35% across different model families. Additionally, as shown in Appendix B.1, TriSpec maintains strong performance across general-domain and extremely challenging benchmarks, further evidencing its ability to generalize beyond standard reasoning tasks.

## 4.3 ROUTING STRATEGIES

We examine two routing families: rule-based and training-based. In the rule-based design, we use only features from the proxy verifier rather than the draft model, because the limited capacity of the draft models can introduce semantic bias into routing decisions. Concretely, we evaluate three rules:

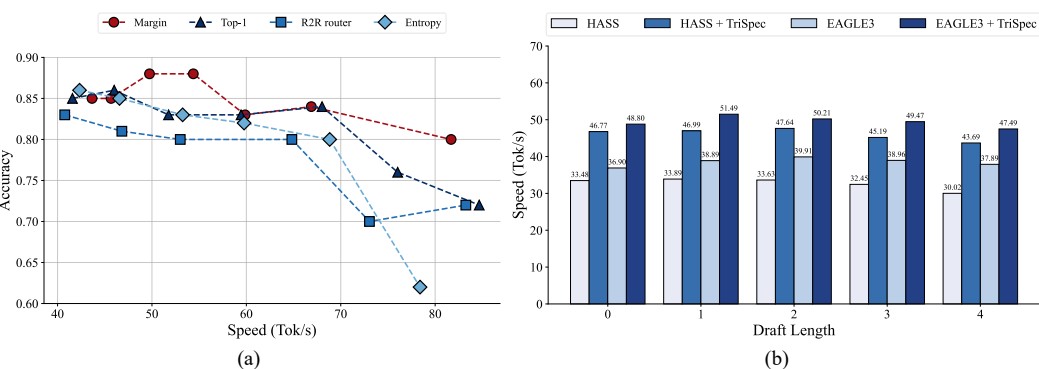

Figure 4: (a) Accuracy–speed trade-offs of different routing strategies. (b) Decoding speed under varying draft lengths for standard speculative decoding and TriSpec. All experiments are conducted on the MATH500 dataset using Qwen3-32B as the target model.

Table 4: Performance under different draft model training strategies on the Qwen3 family. TP denotes Training Parameters in billions.

| Method | Training Strategy | TP | $\tau$ | $r_t$ | Speedup | $\tau$ | $r_t$ | Speedup |
|---|---|---|---|---|---|---|---|---|
| | | | | MATH500 | | | HumanEval | |
| HASS | / | 0.21 | 4.21 | 25.01% | 2.27x | 3.93 | 27.14% | 2.03x |
| HASS + TriSpec | joint training from scratch | 0.26 | 4.39 | 10.24% | 3.18x | 4.42 | 11.61% | 2.99x |
| HASS + TriSpec | adapter-only finetuning | 0.05 | 4.31 | 10.38% | 3.08x | 4.37 | 11.76% | 2.95x |
| EAGLE3 | / | 1.35 | 4.72 | 21.38% | 2.73x | 4.60 | 22.05% | 2.71x |
| EAGLE3 + TriSpec | joint training from scratch | 1.77 | 4.59 | 9.95% | 3.49x | 4.76 | 11.15% | 3.27x |
| EAGLE3 + TriSpec | adapter-only finetuning | 0.42 | 4.52 | 9.99% | 3.44x | 4.65 | 11.40% | 3.21x |

(i) the top-1 probability of the proxy; (ii) a composite criterion that combines top-1 probability and entropy (Wang et al., 2025a); and (iii) the probability margin between the top-1 and top-2 candidates, which is the strategy ultimately adopted by TriSpec. For the training-based approach, we follow the R2R dataset (Fu et al., 2025) and training procedure to train a lightweight router that identifies divergent tokens and coordinates escalation between the proxy and the target.

Figure 4(a) shows the accuracy–speedup trade-offs across routing rules and hyperparameters. Under appropriately calibrated thresholds, all approaches preserve accuracy to within negligible degradation, whereas their attainable speedups vary across methods. Empirically, the probability margin between the top-1 and top-2 candidates provides the strongest accuracy–speed trade-off, so we adopt margin-based routing as the default in TriSpec.

## 4.4 DRAFT MODEL TRAINING STRATEGIES

EAGLE-family draft models are parameter-efficient and relatively inexpensive to train. We recommend joint from-scratch training of the draft model backbone and the adapter, which best accommodates feature inputs from both the target and the proxy. When backbone weights are already available, we also consider adapter-only finetuning for TriSpec to reduce the training cost.

To compare different strategies, we use acceptance length to measure draft quality and training parameters (TP) to assess the training cost. As shown in Table 4, the two strategies yield very similar outcomes: joint from-scratch training attains slightly longer acceptance lengths, while adapter-only finetuning achieves comparable quality with lower training cost. In all cases, the resulting systems deliver end-to-end speed that substantially exceeds standard SD verification.

## 4.5 ABLATION STUDIES.

**Token pruning strategy.** In the case II of TriSpec, the target model participates in the verification process. Since verification is parallelized, the target model can either verify all draft tokens or

trust the proxy verifier's partial validation and verify the remaining tokens, i.e., performing token pruning. Experiments in Table 5 show that the token pruning strategy can increase the acceptance length while maintaining accuracy. This further validates that the proxy's judgments are trustworthy when the proxy exhibits high confidence in its predictions.

**Draft length.** The number of tokens drafted per round (draft length) affects the verification path and the attainable speedup. Under proxy-based verification, TriSpec may accept shorter drafts due to mismatches between proxy and target behaviors, or longer drafts owing to the token pruning mechanism, so the optimal draft length can differ from standard SD. Figure 4(a) compares throughput across draft lengths for standard SD and TriSpec. For the Qwen3 family on MATH500, TriSpec is faster at all draft lengths,

Table 5: Ablation study of token pruning strategy on Qwen3 family. Results are averaged across all five benchmarks.

|  | $\tau$ | Acc. | Speedup |
|---|---|---|---|
| w/ Token pruning | 4.71 | 84.2% | 3.55x |
| w/o Token pruning | 4.58 | 84.2% | 3.46x |

and the optimal draft length shifts modestly relative to standard SD—slightly longer with HASS-trained drafters (from 4 to 5) and slightly shorter with EAGLE3-trained drafters (from 5 to 4).

## 5 RELATED WORK

Beyond the approaches discussed earlier, several lines of work accelerate SD in specific settings. On the drafting side, retrieval-based methods such as REST (He et al., 2023) and LLMA (Yang et al., 2023) identify suitable draft tokens by matching relevant segments from the input context. Cache-reuse approaches, including GLIDE (Du et al., 2024), and MoA (Zimmer et al., 2024), leverage the target model's KV cache for fast drafting. Techniques like layer-skipping (Elhoushi et al., 2024; Zhang et al., 2023) construct simplified draft models by partially reusing the parameters of the target model. Within the single-layer drafter designs, AdaEAGLE (Zhang et al., 2024b) and SpecDec++ (Huang et al., 2024) dynamically control the draft length to avoid unnecessary computation. On the verification side, AASD (Wang et al., 2025a) relaxes checks under controlled conditions, and Traversal Verify adopts a leaf-to-root procedure for sequence-level verification (Weng et al., 2025). Judge Decoding (Bachmann et al., 2025) trains a small judgment model to identify and accept correct but misaligned drafts. Despite these advances, most methods primarily pursue higher acceptance or cheaper drafting; the per-round verification time remains largely unaddressed.

Collaboration between a small language model (SLM) and a large language model (LLM) has been explored for acceleration. R2R (Fu et al., 2025) earns a router to switch between the SLM and LLM. SpecCascade (Narasimhan et al., 2024) combines cascaded inference with speculative decoding to enable cooperative generation. TriForce (Sun et al., 2024) introduces a hierarchical multi-model framework that combines retrieval-based drafting with hierarchical speculation to alleviate KV-cache bottlenecks in long-context settings. However, it is confined to specialized long-context scenarios or requires a drafter with strong capability, making it incompatible with the most efficient single-layer drafter designs. In contrast, we combine an SLM, an LLM, and a single-layer drafter in a ternary framework that leverages the strengths of all three.

## 6 CONCLUSION

In this work, we introduced a new perspective for accelerating speculative decoding by reducing verification time and proposed TriSpec, a ternary framework that combines a lightweight proxy verifier with the drafter and target model. By leveraging same-family smaller models as proxies, TriSpec exploits their strong alignment and reliable outputs to offload a substantial fraction of verification from the target, while a margin-based routing mechanism ensures accuracy is preserved through selective escalation. Extensive experiments show that TriSpec achieves up to 30% additional speedup over standard speculative decoding and sustains accuracy with minimal loss. These results highlight the potential of proxy verification as a new and effective direction for speculative decoding.

## ETHICS STATEMENT

Our work adheres to the ICLR Code of Ethics. This paper introduces TriSpec, a framework for accelerating speculative decoding through proxy-based verification. TriSpec is a methodology-level contribution and does not involve the collection or use of sensitive data, personal information, medical records, or high-risk application domains. All experiments are conducted on publicly available benchmarks for language modeling, reasoning, and code generation, without introducing privacy concerns or potential harm to individuals or communities. We also considered potential societal impacts: TriSpec lowers the computational barrier to LLM deployment, which may broaden accessibility, but could also enable faster misuse if applied to harmful models. We therefore encourage responsible and ethical application of our framework in research and industry.

## REPRODUCIBILITY STATEMENT

We place strong emphasis on the reproducibility of TriSpec. A comprehensive description of the method and experimental design is included in the main text, with further implementation details reported in the Experiment Details section in appendix. All backbone models and benchmark datasets employed in our study are publicly available, allowing researchers to directly replicate our setup. In addition, we document all training and evaluation configurations, including parameter choices, hyperparameter settings, and procedural workflows, to facilitate faithful reproduction of our results. We commit to releasing our repository publicly after the paper's acceptance.

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

## A LLM USAGE STATEMENT

Below we clarify how Large Language Models (LLMs) are used in this study. During manuscript preparation, LLMs served only as broad writing assistants; they did not contribute to idea formation or to drafting the technical substance. In our methodology, LLMs form a central component. Specifically, our approach is designed to accelerate LLM usage; we rely on LLMs (e.g., Qwen3-32B) both for constructing training data and for generating answers used in evaluation. Further specifics appear in the "Experiments" section of the main paper. The authors take full responsibility for all content written under their name.

## B ADDITIONAL EXPERIMENTAL RESULTS

### B.1 PERFORMANCE ON SUPPLEMENTARY BENCHMARKS

To evaluate the effectiveness of our method beyond standard reasoning tasks, we conduct experiments on six additional benchmarks: SpecBench, HotpotQA, Polymath, AIME24, AIME25 and GPQA-diamond. SpecBench comprises 480 samples spanning six categories: multi-turn conversation, translation, summarization, question answering (QA), mathematical reasoning, and retrieval-augmented generation (RAG). Within the multi-turn conversation category, the benchmark includes diverse open-ended tasks such as writing, roleplay, extraction, and humanities. This benchmark offers a comprehensive evaluation of model performance across a wide range of domains. HotpotQA is a natural-language, multi-hop question answering benchmark with explicit supporting-fact supervision, offering a distinct QA format compared to SpecBench, for which we evaluate on a subset of 500 randomly selected samples. Polymath is a multilingual mathematical reasoning benchmark covering 18 languages and four difficulty levels. To assess performance in multilingual and extremely challenging scenarios, we sample 20 high-difficulty instances from each language, yielding 360 test samples in total. AIME-24 and AIME-25 are widely recognized as highly challenging mathematical reasoning benchmarks, with problems drawn from the American Invitational Mathematics Examination. Each dataset consists of 30 competition-level problems. GPQA-Diamond

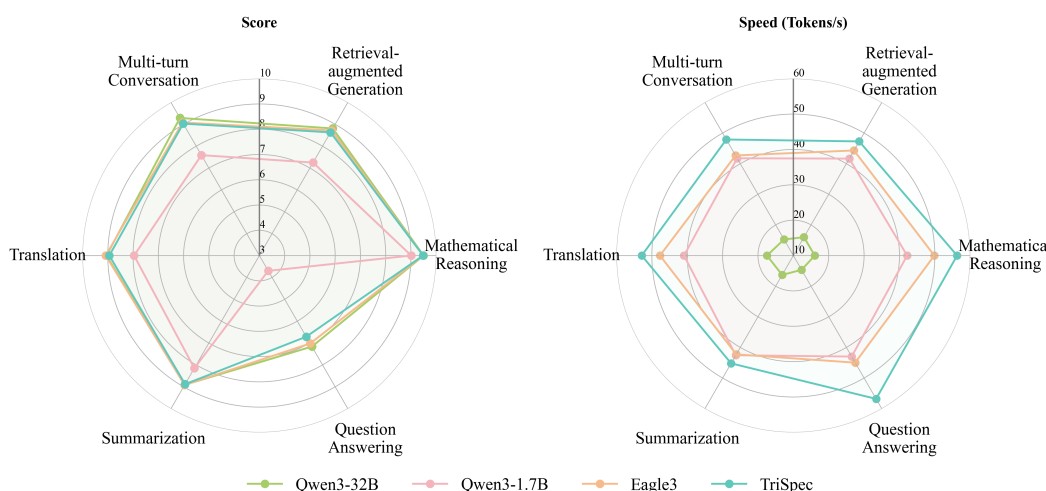

Figure 5: Radar chart illustrating score and speed across all SpecBench categories. All experiments are conducted using models from the Qwen3 family.

is the high-quality subset of the GPQA benchmark, comprising 198 expert-written multiple-choice questions spanning biology, physics, and chemistry. For these datasets, we evaluate on the full test sets to further assess the robustness of our method under extremely challenging problem settings.

For SpecBench, given that most tasks are open-ended, we adopt the LLM-judge protocol from FastChat (Zheng et al., 2023), using GPT-4.1 as the judgment model to score responses from 0 to 10 on a per-turn basis. For HotpotQA and Polymath, we use accuracy as the evaluation metric.

Table 6: Accuracy and speed in general-domain and extremely difficult scenarios.

| | | SpecBench | | HotpotQA | | Polymath | | AIME24 | | AIME25 | | GPQA-Diamond | |
|---|---|---|---|---|---|---|---|---|---|---|---|---|---|
| | Method | Score | Speedup | Acc. | Speedup | Acc. | Speedup | Acc. | Speedup | Acc. | Speedup | Acc. | Speedup |
| Qwen3 | Single Model (Target, 32B) | 8.82 | 1.00x | 93.0% | 1.00x | 27.8% | 1.00x | 24/30 | 1.00x | 22/30 | 1.00x | 67.2% | 1.00x |
| | Single Model (Proxy, 1.7B) | 7.33 | 2.72x | 84.2% | 2.96x | 9.4% | 3.35x | 16/30 | 3.60x | 12/30 | 3.54x | 44.9% | 3.33x |
| | EAGLE3 | 8.77 | 2.80x | 92.8% | 2.77x | 27.20% | 2.34x | 22/30 | 2.52x | 21/30 | 2.31x | 68.2% | 2.97x |
| | EAGLE3+TriSpec | 8.62 | 3.17x | 92.8% | 3.58x | 26.11% | 3.11x | 23/30 | 3.37x | 21/30 | 3.30x | 66.2% | 4.39x |

As shown in Table 6, TriSpec consistently achieves stable speed-up over the standard speculative decoding (SD) framework in both general-domain and extremely difficult tasks. In terms of generation quality, on the SpecBench benchmark, the overall average score decreases by only 0.2 compared to the target model, substantially outperforming the proxy-only setting. On HotpotQA, TriSpec exhibits almost no accuracy degradation. Even on Polymath, AIME24&25 and GPQA-diamond, where the gap between the proxy and the target model is considerable, TriSpec maintains performance close to that of the target model, while delivering more than 30% speed-up over standard SD. Figure 5 presents the score and speed-up for each SpecBench category, further confirming TriSpec's effectiveness across diverse task types.

## B.2 EVALUATION ON FULL BENCHMARKS AND STATISTICAL ROBUSTNESS ANALYSIS

To assess the stability and reliability of our evaluation, we conduct two complementary experiments. First, we extend the evaluation from 100-sample subsets to the full benchmark datasets. Table 7 reports performance and efficiency results on the Qwen3 family models across the complete test sets. Second, we perform repeated evaluation on the GSM8K benchmark by drawing eight distinct random subsets of 100 samples and computing the mean results with error bars, as shown in Figure 6. Error bars in all figures denote one standard deviation across the independent runs.

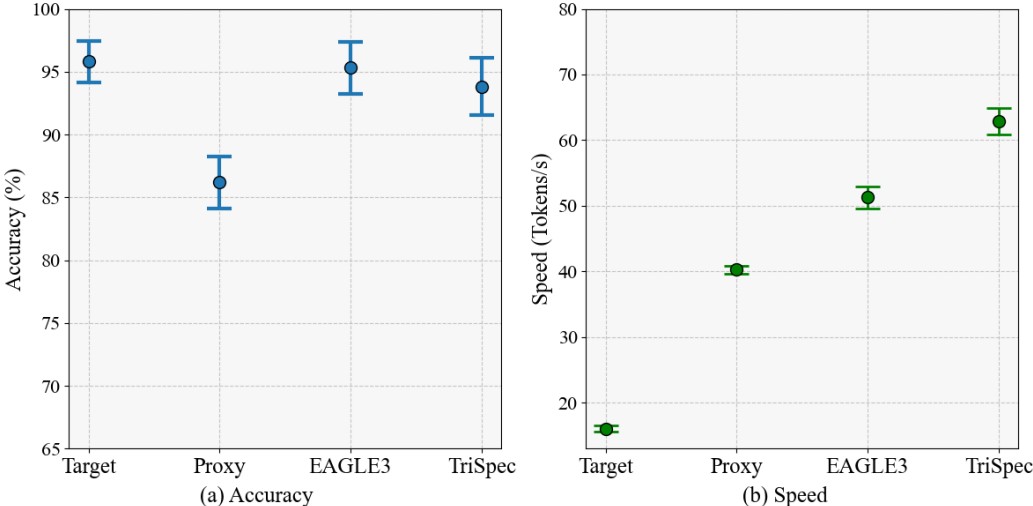

Figure 6: Error bars (one standard deviation) of accuracy and speed for four generation methods across repeated random subset sampling.

Both the full-benchmark evaluation and the repeated-subset experiments consistently show that TriSpec achieves substantially higher generation speed compared to standard speculative decoding (SD), while incurring only minor accuracy degradation. These findings align with our main conclusions and indicate that the observed improvements are not an artifact of subset selection. Moreover, the narrow error bars in Figure 6 indicate that performance variations introduced by random subset sampling are minor, further supporting the robustness of the observed trends.

Table 7: Accuracy and speedup results on the full benchmark suite for Qwen3 family

| | GSM8K | | MATH500 | | Gaokao2023en | | Humaneval | | MBPP | |
|---|---|---|---|---|---|---|---|---|---|---|
| Method | Acc. | Speedup | Acc. | Speedup | Acc. | Speedup | Acc. | Speedup | Acc. | Speedup |
| Single Model (Target, 32B) | 94.9% | 1.00x | 86.8% | 1.00x | 78.2% | 1.00x | 86.6% | 1.00x | 74.8% | 1.00x |
| Single Model (Proxy, 1.7B) | 86.0% | 2.52x | 78.2% | 2.87x | 69.9% | 2.76x | 77.4% | 2.87x | 52.8% | 2.93x |
| EAGLE3 | 94.3% | 3.29x | 88.0% | 2.77x | 77.1% | 2.77x | 87.2% | 2.72x | 74.2% | 2.71x |
| EAGLE3 + TriSpec | 92.9% | 4.08x | 87.4% | 3.48x | 77.4% | 3.39x | 86.0% | 3.23x | 71.4% | 3.33x |

## C  FURTHER ANALYSIS OF THE THREE MODELS IN TRISPEC

### C.1  STANDALONE EXAMPLES

Figure 7 shows a case of individual inferences made by the three models used in TriSpec within the Qwen3 family. The smaller model from the same family (Qwen3-1.7B, proxy verifier) are implicitly aligned with the target model (Qwen3-32B), exhibiting notable similarity in response format and language style, reflecting their strong alignment. Additionally, the smaller model demonstrates reasonable inference capabilities. While there are token-level differences compared to the target model, they are still able to produce correct answers, showcasing the reliability of their outputs.

In contrast, the single-layer drafter (EAGLE3), although trained to mimic the target model's behavior, suffers from weaker contextual understanding due to the significant reduction in attention computations. Without periodic supervision from the target model during generation, the outputs often become illogical.

```
Query:   Calculate 55^2 - 25^2
```

**Qwen3-32B:**
```
We are asked to calculate:
55^2 - 25^2;
This is a difference of squares,
which follows the identity:
a^2 - b^2 = (a - b)(a + b);
Let's apply this identity with
a=55 and b=25:
55^2 - 25^2 = (55 - 25)(55 + 25)
= (30)(80)= 2400;
Final Answer: 2400
```

**Qwen3-1.7B:**
```
We are going to calculate:
55^2 - 25^2;
So we can use the difference of
squares formula:
a^2 - b^2 = (a - b)(a + b);
Let a = 55 and b = 25:
55^2 - 25^2 = (55 - 25)(55 + 25)
= (30)(80) = 2400;
Final Answer: 2400
```

**EAGLE3-Qwen3-32B:**
```
We can use the 55 to 25 to 25 in
the context of a number, and we
can also factorial by 25.
Step1.The number of terms is 1.
Let's say:
- 1/2 (a number of terms) 1.
- 1/2 (a number of terms) 2.
- 1/2 (a number of terms) 3.
    ...
    ...
```

Figure 7: An example of standalone inference for the three models in TriSpec.

## C.2 TOKEN-LEVEL ALIGNMENT EXPERIMENTS

**Experiment details** To compare token-level alignment between the proxy verifier and the target model, we use the ShareGPT dataset as a reference. First, we use Qwen3-32B as the target model to generate autoregressive answers for each prompt. Then, we prefill the target completions using Qwen3-1.7B (the proxy) and record the proxy's next-token prediction at each position. A token is considered an exact match if the proxy and target predictions coincide. For positions where they differ, we consult a stronger judge (Qwen3-235B) with top-k = 3 and top-p = 0.9 sampling parameters: if the proxy's token is included in the set of tokens generated under these conditions, we deem it acceptable; otherwise, it is considered unacceptable. For both the proxy and the target model, temperature = 0 is used to ensure deterministic outputs.

**Case Analysis** We also performed token-level alignment comparisons for the single-layer drafter (EAGLE3) and present a case in Figure 8, showcasing the token-level outputs of the three models along with their corresponding confidence scores. The proxy verifier typically shows higher confidence for tokens that are exact matches, and slightly lower confidence for mismatched tokens, demonstrating the reliability of its judgments. Overall, the proxy verifier outperforms the EAGLE3 drafter in alignment (82% vs. 68%). This advantage is particularly evident for decision-critical tokens (e.g., "*" and "78"), enabling the proxy verifier to locally correct errors made by the single-layer drafter in certain cases. Additionally, due to its limited capacity, the single-layer drafter exhibits overconfidence in certain cases, making its confidence scores unreliable as a basis for determining whether its outputs are acceptable.

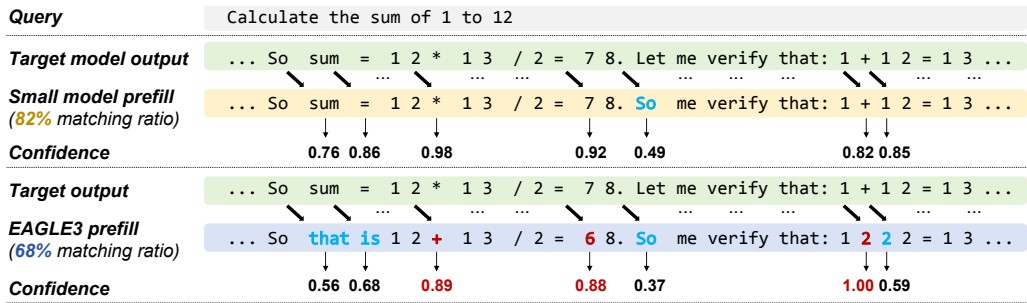

Figure 8: Token-level comparison of three models in TriSpec. Blue highlights mark acceptable mismatch tokens, and red highlights mark unacceptable mismatches. The reported matching ratios are measured on all ShareGPT samples.

## C.3 ATTEMPTS AT RELAXING VERIFICATION

This work primarily focuses on the verification side, aiming to accelerate SD by relaxing strict verification constraints. In addition to the proxy-based verification used in TriSpec, another natural approach is to relax verification conditions to increase acceptance length, thereby achieving acceleration. We examine two common relaxations in the SD pipeline: (i) confidence filtering,

which accepts a drafter token when the drafter's confidence score ($\max p_i$) exceeds a threshold and skips target verification, and (ii) top-k verification, which accepts the drafter token if it lies within the top-k candidates of the target model. We experiment by separately using single-layer drafters (HASS/EAGLE3 weights) and same-family smaller models as draft models. In these experiments, we use a chain-based draft structure, with the draft length set to 3.

**Single-layer drafter is difficult to relax.** As shown in Figure 9(b), single-layer drafters degrade rapidly under both settings. Confidence filtering admits high-confidence but incorrect drafts because the single-layer drafter is capacity-limited and often overconfident. Top-k verification is brittle, as it can accept incorrect near-optimal tokens, which single-layer drafters frequently emit at decision-critical positions. Consequently, for single-layer drafters, relaxed verification is fragile, and small relaxations already induce noticeable accuracy loss, which limits further acceleration.

In our benchmark evaluation, we comprehensively assess the confidence-filter method across multiple benchmarks, with results reported in Table 1. Despite setting a relatively high threshold of 0.95, the relaxed verification based on the confidence filter still struggles to maintain accuracy. In particular, it frequently induces repeated outputs, which we attribute to the single-layer drafter's unwarranted overconfidence discussed in Appendix C.2. This is also why we do not rely on feature signals from the draft model when designing the routing strategy.

**High-confidence mismatches in same-family smaller models can be tolerated.** In contrast, relaxing verification for smaller models from the same family, when using an appropriate threshold, does not lead to significant accuracy loss. This end-to-end experiment shows that high-confidence mismatches in smaller models can typically be tolerated, further supporting their rationale as proxy verifiers. A similar conclusion is also mentioned in the Judge Decoding (Bachmann et al., 2025).

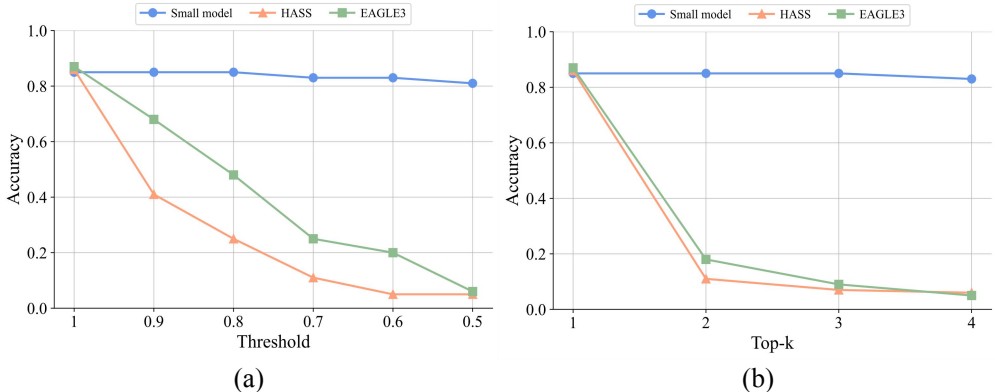

(a)                                      (b)

Figure 9: Verification relaxation in the Qwen3 family. The target model is Qwen3-32B, the small model is Qwen3-1.7B, and HASS/EAGLE3 denote single-layer drafters trained with the respective methods. (a–b) Accuracy on MATH500 under relaxed verification (confidence filtering and top-k verification) across different drafters.

## C.4 PROXY SELECTION IN TRISPEC

**Token-Level Alignment across Model Scales** Figure 10 compares the token-level exact match ratio between Qwen3 models of varying parameter counts and the Qwen3-32B target model. Match ratios increase with model size, but the marginal gains diminish beyond the 1.7B scale. Since token-level alignment directly impacts a proxy's ability to assist the target in verification, this trend is a key factor in proxy selection. In particular, the 0.6B model surpasses the single-layer EAGLE3 drafter by only about 6%, yielding relatively few local corrections. Conversely, the 4B model offers merely a 2.5% increase over the 1.7B model, yet incurs substantially higher inference overhead. Balancing speed and alignment effectiveness, we adopt the 1.7B model as the proxy in our main experiments.

**Proxies from Different Model Families**   Employing cross-family models as proxies is a possible alternative, but it faces fundamental compatibility constraints. In particular, incompatible vocabularies and tokenizers hinder reliable token-level verification and accurate local corrections of drafted tokens. Furthermore, proxies whose generation behavior diverges substantially from that of the target tend to bias verification decisions toward their own output distribution, lowering acceptance rates and disrupting the cooperative dynamics between the draft and target model. For these reasons, TriSpec employs same-family proxies, which share the tokenizer and closely match the target's output distribution, thereby enabling consistent token-level verification and maintaining stable accuracy-latency trade-offs.

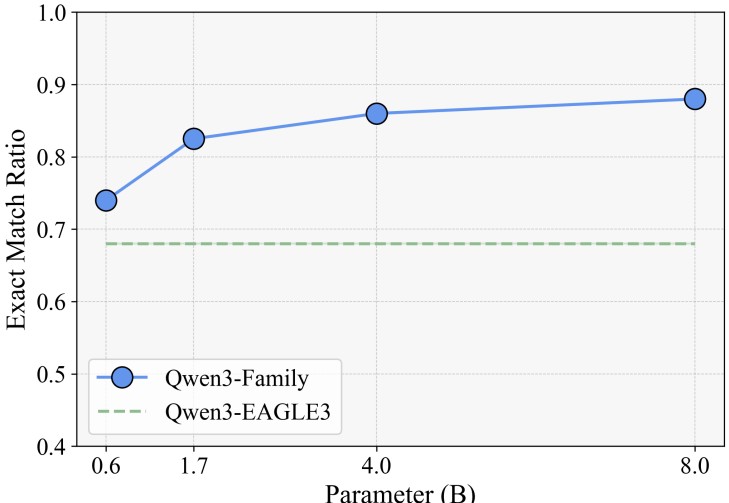

Figure 10: Token-level exact match ratios of models with varying parameter scales and the EAGLE3 drafter, computed using Qwen3-32B predictions as reference.

## D   EXPERIMENT DETAILS

### D.1   DRAFT TRAINING DETAILS

We follow the official codebases and guidelines from the respective papers to train the draft models. In TriSpec, the drafter's backbone and adapter are jointly trained from scratch. Specifically, HASS is trained on 68K ShareGPT examples, and EAGLE3 additionally incorporates UltraChat-200K (Ding et al., 2023). Concretely, we first generate responses autoregressively with the target model and then compute token-level hidden features offline with both the proxy and the target to supervise baseline training and our adapter. Training is performed on $8\times$ NVIDIA A100 80GB GPUs, with hyperparameters set according to the official default settings unless otherwise noted. Notably, during the EAGLE3 training process, we maintain a full vocabulary for training and set the batch size to 1 to avoid out-of-memory errors. In terms of training time, for the 32B target model, HASS/EAGLE3 typically requires 3-4 days, while TriSpec takes approximately 4-5 days due to the additional requirement of training the proxy's features. The details of the training cost are shown in Table 8.

During inference, we follow the scripts provided by the official EAGLE implementation. All benchmark inference experiments are conducted on a single NVIDIA A100 80GB GPU.

Table 8: Training cost of EAGLE3 and TriSPec on Qwen3 family

| Method | Training Strategy | Data Size | Seq Len | TP(B) | Epochs | Training Time(h) |
|---|---|---|---|---|---|---|
| EAGLE3 | / | 518K | 2048 | 1.35 | 6 | 78 |
| EAGLE3 + TriSpec | joint-training | 518K | 2048 | 1.77 | 6 | 115 |
| EAGLE3 + TriSpec | adapter-only | 518K | 2048 | 0.42 | 3 | 44 |

## D.2 BASELINE IMPLEMENTATION DETAILS: SPECCASCADE

We adopt SPECCASCADE as one of our main baselines. SPECCASCADE encompasses various rule-based strategies for relaxed verification within speculative decoding. In our experiments, we implement three representative *deferral rules*:

- **Chow rule**: This method compares the probability assigned by the draft model to a candidate token against a fixed threshold, accepting the token if it exceeds this threshold (see Table 1 in the original SpecCascade paper). Notably, our *confidence-filter* method described in Appendix C.2 is equivalent to the Chow rule.

- **Optimal deferral rule**: This method compares the probability assigned by the draft model to the maximum probability from the target model's distribution. The rejection threshold in this method is determined as a function of the distance between the two distributions (see Eq. (10) in the original paper).

- **Token-specific deferral rule**: This method decides acceptance by comparing the probability of the candidate token in the target model's distribution against the target's maximum probability. It does not use the draft model's output distribution at all (see Eq. (14) in the original paper).

In our implementation, we set a small $\alpha = 0.05$ for rules relying on the draft model's output distribution (Chow and Optimal deferral), and a relatively larger $\alpha = 0.2$ for the token-specific deferral rule.

As shown in Table 1 of the revised paper, when the draft model is a *small single-layer* architecture, deferral rules that rely on the draft model's probability severely harm accuracy: even with very small $\alpha$, Chow and Optimal deferral rules still incur substantial degradation, consistent with our findings in Appendix C.2. In contrast, the token-specific deferral rule maintains accuracy by relying solely on the target model's distribution. However, because the small drafter can only produce a limited number of high-quality tokens, the resulting speedup gains from this rule are modest.

## E CASE STUDY OF COMPLETE GENERATION

We present a TriSpec case study of complete generation and compare it with target-only and proxy-only decoding results(Figure 11). Tokens in blue denote outputs verified or generated by the proxy model (DSQ-1.5B), while tokens in black indicate those verified or generated by the target model (DSQ-32B). In this example, over half of the tokens are validated by the proxy model, highlighting TriSpec's efficiency. When the proxy alone would err, TriSpec attains correctness by involving the target, preserving accuracy.

Specifically, the yellow boxes highlight disagreements between the two models. The proxy incorrectly predicts "He runs 3 sprints each week" instead of the correct "He runs 3 sprints each session." In TriSpec inference, however, the proxy expresses uncertainty at this point and defers to the target model, thereby avoiding the error and showcasing strong robustness.

**Case: GSM8K**

**Question:** James decides to run 3 sprints 3 times a week. He runs 60 meters each sprint. How many total meters does he run a week?

**GT:** 540

---

**DSQ-32B**

First, I need to determine the total number of sprints James runs each week. He runs 3 sprints each session and does this 3 times a week. So, the total number of sprints is 3 multiplied by 3, which equals 9 sprints.

Next, I need to calculate the total distance James runs. Each sprint is 60 meters, and he completes 9 sprints in total. Therefore, the total distance is 60 meters multiplied by 9, which equals 540 meters.

Finally, the total number of meters James runs each week is 540 meters.

Solution:

To determine the total number of meters James runs in a week, follow these steps:

1. Calculate the total number of sprints per week:

James runs 3 sprints each session and does this 3 times a week.

Total sprints = 3 sprints/session × 3 sessions/week = 9 sprints/week

2. Determine the total distance run per week:

Each sprint is 60 meters. Therefore, the total distance is:

Total distance = 9 sprints × 60 meters/sprint = 540 meters

Final Answer:

540

---

**DSQ-1.5B**

I need to determine how many total meters James runs in a week. He runs 3 sprints each week, and each sprint is 60 meters long. To find the total distance, I'll multiply the number of sprints by the distance per sprint.

So, 3 sprints multiplied by 60 meters per sprint equals 180 meters. Therefore, James runs a total of 180 meters each week.

James runs 3 sprints each week, with each sprint being 60 meters long. To find the total distance he runs in a week, multiply the number of sprints by the distance per sprint.

Total meters = Number of sprints × Distance per sprint = 3 × 60 = 180

180

---

***TriSpec***

First, I need to determine the total number of sprints James runs in a week. He runs 3 sprints each session and does this 3 times a week. So, the total number of sprints is 3 multiplied by 3, which equals 9 sprints.

Next, each sprint is 60 meters long. To find the total distance James runs in a week, I multiply the number of sprints by the distance of each sprint. That is 9 sprints multiplied by 60 meters per sprint, resulting in 540 meters.

Therefore, James runs a total of 540 meters each week.

Solution:

To determine the total number of meters James runs in a week, follow these steps:

1. Calculate the total number of sprints per week:

James runs 3 sprints each session and does this 3 times a week.

Total sprints per week = 3 sprints/session × 3 sessions/week = 9 sprints/week

2. Determine the total distance run per sprint:

James runs 60 meters per sprint

3. Calculate the total distance run in a week:

Multiply the total number of sprints by the distance per sprint.

Total distance = 9 sprints × 60 meters/sprint = 540 meters

Final Answer:

540

Figure 11: Case study of TriSpec in GSM8K.

