# OpenReview forum: "TriSpec: Ternary Speculative Decoding via Lightweight Proxy Verification"
_ICLR.cc/2026/Conference — Submitted to ICLR 2026_

### Official Review · Reviewer_EJNW · 2025-10-20

**Soundness:** 2
**Presentation:** 2
**Contribution:** 2
**Rating:** 4
**Confidence:** 4

**Summary:**

The paper proposes TriSpec, a ternary speculative decoding (SD) framework that adds a lightweight proxy verifier between the drafter and the target model to reduce verification cost. A margin-based routing rule decides when the proxy's verification is "trusted". Several experiments on Qwen3 and DeepSeek-R1-Distill-Qwen demonstrate the effectiveness of the method.

**Strengths:**

- This paper is overall well-written.

- The idea of applying a proxy verification model offers a new angle on SD efficiency to reduce the verification time

- Several experiments demonstrate the effectiveness of the TriSpec.

**Weaknesses:**

- Results are limited to two families (Qwen3, DSQ). It’s unclear how well the “same-family small proxy” assumption holds for other backbones, including Llama 2, Llama 3, and the Vicuna series.
- Accuracy is measured via pass@1 on math/code; there’s little analysis of generation fidelity for open-ended text or long-form reasoning where subtle proxy deviations could matter.
- TRISPEC itself cannot strictly speed up the LLM reasoning process losslessly. Some minor differences are unacceptable in some fields, such as medicine and law.
- Trispec adds a proxy model, which also brings additional deployment and memory overhead。

**Questions:**

Please refer to weakness.

---

> ### Author Response · Authors · 2025-11-21
> **Response by Authors to EJNW (1/3)**
>
> > **W1**: Results are limited to two families (Qwen3, DSQ). It's unclear how well the "same-family small proxy" assumption holds for other backbones, including Llama 2, Llama 3, and the Vicuna series.
>
> We appreciate the reviewer's concern about generality. To explicitly test whether the effectiveness of TriSpec depends on the Qwen architecture, we added experiments using DeepSeek-R1-Distill-LLaMA, which is based on the LLaMA family and therefore architecturally distinct from Qwen. We evaluate TriSpec with a 70B target model and an 8B same-family proxy verifier.
>
> As shown in the tables below (corresponding to Table 1 and Table 3 in the revised paper), the results demonstrate that:
>
> 1. A **3.36× speedup** over target-only decoding, and approximately **40 % additional speedup** over standard speculative decoding on average.
> 2. Approximately **40%-60% reduction** in target model invocations
> 3. **Less than 2%** accuracy degradation in average accuracy across all five benchmarks
>
> These results provide concrete evidence that TriSpec's proxy-verifier mechanism is **architecture-agnostic** and that smaller model variants can serve as effective verifiers beyond a single model provider.
>
> |Method|GSM8K Acc.|GSM8K Speedup|MATH500 Acc.|MATH500 Speedup|GK23-en Acc.|GK23-en Speedup|HumanEval Acc.|HumanEval Speedup|MBPP Acc.|MBPP Speedup|Avg Acc.|Avg Speedup |
> |-|-|-|-|-|-|-|-|-|-|-|-|-|
> |Single Model (Target,70B)|88%|1.00x|90%|1.00x|78%|1.00x|87%|1.00x|79%|1.00x|84.4%|1.00x |
> |Single Model (Proxy,8B)|77%|3.71x|83%|3.73x|73%|4.01x|80%|3.89x|59%|4.03x|74.4%|3.87x |
> |EAGLE3|88%|2.83x|89%|2.35x|76%|2.31x|84%|2.49x|80%|2.40x|83.4%|2.48x |
> |EAGLE3 + SC[Chow]|82%|2.90x|81%|2.50x|65%|2.61x|74%|2.58x|71%|2.57x|74.6%|2.63x |
> |EAGLE3 + SC[OPT]|78%|3.02x|66%|2.84x|48%|2.93x|60%|2.89x|51%|2.91x|60.6%|2.92x |
> |EAGLE3 + SC[Token]|87%|2.92x|88%|2.43x|75%|2.56x|85%|2.69x|78%|2.65x|82.6%|2.65x |
> |EAGLE3 + TriSpec|86%|**3.74x**|88%|**3.35x**|77%|**3.36x**|87%|**3.07x**|76%|**3.29x**|82.8%|**3.36x** |
>
> |Method|MATH500 $r_t$|MATH500 $t_v$(s)|MATH500 $\tau$|MATH500 $L$(s)|MATH500 Speed(Tok/s)|HumanEval $r_t$|HumanEval $t_v$(s)|HumanEval $\tau$|HumanEval $L$(s)|HumanEval Speed(Tok/s)|
> |-|-|-|-|-|-|-|-|-|-|-|
> |Single Model (Target,70B)|100.00%|/|/|256.79|8.22|100.00%|/|/|254.18|8.37 |
> |EAGLE3|28.70%|0.144|3.48|108.89|19.33|26.88%|0.147|**3.66**|107.92|20.84 |
> |EAGLE3 + TriSpec|**11.60%**|**0.093**|**3.64**|**78.32**|**27.54**|**15.18%**|**0.114**|3.59|**87.51**|**25.70** |

---

> ### Author Response · Authors · 2025-11-21
> **Response by Authors to EJNW (2/3)**
>
> > **W2**: Accuracy is measured via pass@1 on math/code; there’s little analysis of generation fidelity for open-ended text or long-form reasoning where subtle proxy deviations could matter.
>
> Thank you for the suggestion. We address this concern in two ways:
> 1. **Beyond math/code (more general domains)**
>    We extend our evaluation to:
>    - **SpecBench**, which spans six categories and a wide range of fine‑grained tasks, including but not limited to question answering,  translation, writing, roleplay, and summarization, offering a comprehensive measure of open-ended generation quality.
>    - **HotpotQA**, a natural-language, multi-hop question answering benchmark with explicit supporting-fact supervision, offering a distinct style of QA evaluation compared to the tasks in SpecBench.
>
>    As shown in the table below, which corresponds to Table 6 in the revised paper:
>    - **SpecBench**: TriSpec achieves a **3.17×** speedup over target‑only decoding and **13%** faster than the standard speculative decoding baseline, while the overall average score decreases by only **0.2** compared to the target model.
>    - **HotpotQA**: TriSpec achieves a **3.58×** speedup over the target model and an additional **29%** speedup compared to the standard speculative decoding baseline, while maintaining essentially the same accuracy as the target.
>
>    In addition, **Figure 5** in the revised paper presents TriSpec’s performance across the **individual task categories** within SpecBench, demonstrating its effectiveness across all task types.
> 2. **Long-form reasoning in high-gap regimes**
>    We further explore TriSpec’s capability on long‑form reasoning by evaluating **Polymath** at the high‑difficulty level, a setting that demands numerous reasoning steps and complex intermediate inference to reach the final answer. In this regime, the proxy model’s accuracy is roughly one‑third of the target’s, yet TriSpec recovers about **95%** of the target’s accuracy while achieving a **3.11× speedup**, highlighting its robustness even under substantially weaker proxy performance in highly challenging reasoning tasks.
>
> These results demonstrate that TriSpec maintains high generation fidelity across a wide range of tasks, from open‑ended text generation to complex long‑form reasoning, indicating strong generalization across diverse scenarios.
>
> | Model | Method | SpecBench Score | SpecBench Speedup | Polymath Accuracy | Polymath Speedup | HotpotQA Accuracy | HotpotQA Speedup |
> |-------|--------|-----------------|-------------------|-------------------|------------------|-------------------|------------------|
> | Qwen3| Single Model (Target, 32B) | 8.82 | 1.00x | 27.8% | 1.00x | 93.0 | 1.00x |
> |       | Single Model (Proxy, 1.5B) | 7.33 | 2.72x | 9.4% | 3.35x | 84.2 | 2.96x |
> |  | EAGLE3 | 8.77 | 2.80x | 27.20% | 2.34x | 92.8 | 2.77x |
> |       | EAGLE3+TriSpec | 8.62 | 3.17x | 26.11% | 3.11x | 92.8 | 3.58x |

---

> ### Author Response · Authors · 2025-11-21
> **Response by Authors to EJNW (3/3)**
>
> > **W3**: TRISPEC itself cannot strictly speed up the LLM reasoning process losslessly. Some minor differences are unacceptable in some fields, such as medicine and law.
>
> Thank you for raising this important point. We agree that in safety‑critical domains such as medicine or law, even minor deviations may be unacceptable. TriSpec is designed with this in mind and provides an improved trade‑off between speed and accuracy:
>
> 1. **Controllable verification margin**
>    TriSpec allows users to set a margin‑based threshold to control the proportion of proxy usage. In high‑precision scenarios, users can increase the threshold to reduce reliance on the proxy; setting it to 1.0 completely disables proxy‑based verification and reverts to the lossless standard speculative decoding framework. For non‑critical workloads (e.g., casual dialogue, drafting, brainstorming), the threshold can be lowered to increase speed, and can even be set to 0.0 to generate entirely with the proxy model for maximum throughput.
> 2. **Precedent for accuracy–speed trade‑offs**
>    Prior verification‑side methods (e.g., Judge Decoding[1], SpecCascade[2]) have also traded small amounts of accuracy for substantial speed gains. TriSpec adopts this proven paradigm, with the added flexibility to revert instantly to a fully lossless speculative decoding mode when needed.
> 2. **Mitigation in sensitive domains**
>    In practice, even the target LLM can produce occasional errors in safety‑critical applications. Such domains often employ safeguards such as retrieval‑augmented generation (RAG) or external fact‑checking to ensure correctness. The proxy verifier is also capable of using these external signals when making its judgments, rather than relying solely on its own parametric knowledge. This makes TriSpec suitable for cautious application in sensitive domains.
>
> In summary, TriSpec can operate in a fully lossless standard speculative decoding mode when absolute precision is required, ensuring no degradation in reasoning accuracy. In settings where slight deviations are acceptable, it delivers substantial speed‑ups over the lossless baseline, providing both reliability for safety‑critical domains and efficiency for general applications.
>
> [1] Bachmann, Gregor, Sotiris Anagnostidis, Albert Pumarola, Markos Georgopoulos, Artsiom Sanakoyeu, Yuming Du, Edgar Schönfeld, Ali Thabet, and Jonas Kohler. "Judge decoding: Faster speculative sampling requires going beyond model alignment." arXiv preprint arXiv:2501.19309 (2025).
>
> [2] Narasimhan, Harikrishna, Wittawat Jitkrittum, Ankit Singh Rawat, Seungyeon Kim, Neha Gupta, Aditya Krishna Menon, and Sanjiv Kumar. "Faster cascades via speculative decoding." arXiv preprint arXiv:2405.19261 (2024).
>
> > **W4**: Trispec adds a proxy model, which also brings additional deployment and memory overhead
>
> Thank you for highlighting this concern. While introducing a proxy model does incur extra memory usage, the overhead is modest because the proxy is typically much smaller than the target model. For example, when using Eagle3 as the draft model and Qwen3‑32B as the target, with a pre‑allocated KV cache length of 12k, single‑GPU inference requires about 73.15 GB of memory in the standard speculative setting. Under the same configuration, adding a 1.7B‑parameter proxy model for TriSpec increases the requirement by only **5.20 GB** (**~7%** extra memory usage). We consider this acceptable, as such “space‑for‑time” trade‑offs are common in high‑performance inference and do not form a practical bottleneck for deployment.

---

> > ### Comment · Reviewer_EJNW · 2025-11-26
> >
> > I thank the authors for their detailed response. However, I find that my concerns have not been adequately addressed.
> >
> > - The Non-Lossless Issue: One of the most appealing features of standard Speculative Decoding methods is their theoretically lossless nature. This guarantees that SD can always serve as a "last-mile" solution to accelerate downstream tasks without altering the target model's output distribution. While TriSpec's introduction of a proxy model improves efficiency, it sacrifices this lossless characteristic. In my view, this trade-off significantly limits the scope of the method in practical applications and remains a notable concern.
> >
> > - The Deployment Issue: TriSpec's tri-layer architecture is currently unsupported by mainstream inference frameworks (e.g., vLLM, sglang). Furthermore, it cannot be straightforwardly deployed within a prefill-decoding disaggregation architecture, which reduces its feasibility for real-world deployment. Additionally, enabling Tensor Parallelism on a single node for this three-layer structure faces inherent restrictions, and implementing interactions across multiple communication groups presents significant engineering challenges.
> >
> > Consequently, I believe TriSpec requires major revisions in its current form to address these practicality gaps. Therefore, I am inclined to maintain my current negative score.

---

> ### Author Response · Authors · 2025-11-26
>
> We thank the reviewer for raising these important points regarding both the lossless property and the deployment feasibility of TriSpec. We will present our perspective on each concern in turn.
>
> On the lossless property, we note that non‑lossless acceleration methods have already been widely adopted in LLM inference and form an established research line. Approaches such as relaxed verification in speculative decoding [1,2,3,4] and collaborative inference between large and small models [5] intentionally trade strict fidelity to the target model’s outputs for substantial speed‑ups. In this context, empirical results across diverse benchmarks show that TriSpec’s accuracy degradation is no more than 3% relative to the target model, while achieving markedly better speedup than existing relaxed‑verification methods (e.g., SpecCascade in Table 1). These findings underline TriSpec’s practicality for real‑world deployment. Additionally, we emphasize that TriSpec is designed to provide a configurable accuracy–efficiency trade‑off: for scenarios where strict lossless behaviour is critical, it can be run in a conservative mode (e.g., applying a high‑margin threshold or falling back entirely to the original speculative decoding), thereby preserving the target model’s output distribution.
>
> On the deployment issue, we believe that TriSpec’s architecture can be adapted to mainstream inference frameworks with modest engineering effort. In a prefill–decoding disaggregation setting, the proxy verifier, draft model, and target model each build their own KV caches within the prefill instance during the prefill stage. During the subsequent decoding stage, the proxy verification mechanism is executed within the decode instances: the draft model generates candidate tokens, the proxy verifier performs intermediate acceptance checks, and any uncertain tokens fall back to verification by the target model. This preserves the established separation between prefill instances and decode instances, and the detailed procedure can be made consistent with the standard speculative decoding setup, such as the **[EAGLE3 pipeline in SGLang](https://docs.sglang.io/advanced_features/speculative_decoding.html)**. On the tensor parallelism design, TriSpec can also mirror the communication pattern of standard speculative decoding: in standard SD, TP communication occurs solely within each model’s respective TP group, and inter‑model exchanges are managed at the controller level rather than via cross‑group GPU‑to‑GPU tensor transfers. TriSpec follows this same principle: only the draft interacts with the proxy or the target, enabling the proxy+draft and target+draft stages to be implemented using two separate communication schemes based on the standard SD pattern. Consequently, each model maintains its own TP group, and inter‑model coordination is fully managed in control logic, thereby avoiding the complexity and overhead associated with cross‑group tensor communication.
>
> In conclusion, TriSpec provides an efficient accuracy–speed trade‑off strategy, offering a new option for practical application scenarios. Regarding deployment, we believe that TriSpec is not inherently incompatible with mainstream inference frameworks, and the related optimization schemes will be provided in subsequent versions.
>
> [1] Bachmann G, Anagnostidis S, Pumarola A, et al. Judge decoding: Faster speculative sampling requires going beyond model alignment. (ICLR 2025)
>
> [2] Narasimhan H, Jitkrittum W, Rawat A S, et al. Faster cascades via speculative decoding. (ICLR 2025)
>
> [3] Qin Z, Hu Z, He Z, et al. Optimized multi-token joint decoding with auxiliary model for llm inference. (ICLR 2025)
>
> [4] Liao B, Xu Y, Dong H, et al. Reward-guided speculative decoding for efficient llm reasoning. (ICML 2025)
>
> [5] Fu T, Ge Y, You Y, et al. R2R: Efficiently Navigating Divergent Reasoning Paths with Small-Large Model Token Routing. (NIPS 2025)

---

### Official Review · Reviewer_iPrA · 2025-10-27

**Soundness:** 2
**Presentation:** 3
**Contribution:** 2
**Rating:** 4
**Confidence:** 4

**Summary:**

The paper proposes **TriSpec**, a ternary speculative decoding (SD) pipeline that inserts a **lightweight proxy verifier** between the usual drafter and the target LLM. The proxy is a smaller, same‑family model (e.g., Qwen3‑1.7B for Qwen3‑32B) that pre‑verifies drafted tokens and locally corrects the first rejection; the expensive target model is called only when the proxy’s **margin test** (top‑1 minus top‑2 probability) marks positions as untrusted. The authors extend EAGLE‑style single‑layer drafters with a small **adapter** so the drafter can be seeded by proxy or target features. Algorithm 1 and Fig. 1–3 describe the flow; two regimes are covered: proxy completes the round without target, or the target verifies the untrusted suffix with **token pruning** of proxy‑trusted tokens. Experiments on Qwen3 and DeepSeek‑R1‑Distill‑Qwen show up to ~30% higher speedup than standard SD pipelines (HASS/EAGLE‑3) at ≤1% average accuracy loss, with >50% fewer target invocations.

**Strengths:**

1. The paper identifies verification time as a first-order bottleneck in modern SD stacks and operationalizes a clear, reproducible fix: insert a same-family proxy and gate escalation with a top-1 vs. top-2 margin. The algorithm is simple to implement atop EAGLE-family drafters.
2. The presentation and the figures are intuitive and easy to understand.
3. The experiments show large reductions in target-invocation ratio and lower per-round verification time, while keeping acceptance length stable.

**Weaknesses:**

1. Novelty is limited versus recent verification-side work. While the motivation to reduce target calls with a cheaper verifier is straightforward, the idea of introducing a mid-level LLM into the draft model and the target model is well explored.
2. TriSpec achieves better speedup ratio at the cost of losing the theoretical lossless property of speculative decoding, which is especially important in real-world applications. The method can accept proxy-approved tokens that differ from the target. While **Appendix B** argues these are usually acceptable, there is no stress test on open-ended generation, multilingual prompts, or safety-sensitive settings where small token changes may carry large semantic shifts. Meanwhile, The paper should quantify quality shifts under temperature and across domains, not only average accuracy.
3. Missing Related Works. Some related works [1, 2] already explored the idea of multi-level speculative decoding. The lack of these baselines weakens the novelty and evidence.
4. Evaluation is narrow and controlled. The experiments are only conducted on 2 Qwen-32B series models. The effectiveness of TriSpec on large-scale LLMs (>=70B) and other LLM backbones (e.g. llama and GLM) remains unknown.

[1] Bachmann, Gregor, Sotiris Anagnostidis, Albert Pumarola, Markos Georgopoulos, Artsiom Sanakoyeu, Yuming Du, Edgar Schönfeld, Ali Thabet, and Jonas Kohler. "Judge decoding: Faster speculative sampling requires going beyond model alignment." *arXiv preprint arXiv:2501.19309* (2025).

[2] Narasimhan, Harikrishna, Wittawat Jitkrittum, Ankit Singh Rawat, Seungyeon Kim, Neha Gupta, Aditya Krishna Menon, and Sanjiv Kumar. "Faster cascades via speculative decoding." *arXiv preprint arXiv:2405.19261* (2024).

**Questions:**

1. Could you please specify the detailed training cost of the draft model?
2. Could you please provide more experiments on some extremely difficult tasks? Will TriSpec significantly decrease the model performance? If the user's query is out of the domain of the training data of the proxy model, may the proxy model give low-quality judge?
3. Does the proxy and the target model run on the same GPU? Whether the KV cache is shared?

---

> ### Author Response · Authors · 2025-11-21
> **Response by Authors to iPrA (1/4)**
>
> > **W1**: Novelty is limited versus recent verification-side work. While the motivation to reduce target calls with a cheaper verifier is straightforward, the idea of introducing a mid-level LLM into the draft model and the target model is well explored.
>
> We thank the reviewer for raising this point. While recent verification‑side work (e.g., Judge Decoding, SpecCascade) indeed introduces intermediate components between the draft model and the target model, and explores various strategies for relaxing verification constraints, our approach differs substantially in **motivation**, **optimization objective**, and **compatibility with current state‑of‑the‑art drafters**, while also addressing some inherent limitations of prior methods.
>
> 1. **Different motivation and optimization objective**
>    Most prior verification‑side methods (e.g., Judge Decoding, SpecCascade) aim to relax the strict verification rules in the standard speculative decoding framework so that certain draft tokens can still be accepted even if their probability distribution does not exactly match the target's. The central goal in these methods is to increase the average acceptance length $\tau $ in Eq. (3).
>    In contrast, TriSpec focuses on reducing the per‑round verification cost $t_v$ by delegating part of the target's verification work to a lightweight, same‑family proxy verifier. This is a different optimization target and does not conflict with acceptance‑rate‑based strategies.
> 2. **Addressing limitations of prior methods**
>    Acceptance‑rate‑based approaches often require the drafter to have sufficiently strong capability; if the drafter is not high‑quality, the number of usable tokens is limited, and unreliable output distributions can mislead relaxed‑verification rules. However, the most efficient and widely adopted drafters today, such as **EAGLE3** and **MTP**, are extremely small single‑layer models trained to align with the target's outputs. These achieve state‑of‑the‑art speedup, but offer relatively weak standalone performance.
>
>    In practice, prior methods often fail to retain their original benefits when applied to such small single‑layer drafters:
>    - Judge Decoding explicitly states in the limitations section of its appendix: **"The draft model needs to be of high quality, ... small drafters in the spirit of Medusa or Eagle are thus not ideal."**
>    - Our experiments for SpecCascade in Table 1 show that with a single‑layer drafter, simple rules such as Chow or optimal deferral fail to maintain accuracy, due to unreliable drafter probabilities. Token‑specific deferral rules can preserve accuracy but provide only modest speed gain.
>    - Our trials in **Appendix C.3** confirm that single‑layer drafters cause several commonly used relaxation strategies to break down.
>
>    TriSpec avoids this limitation by targeting the verification‑stage cost rather than weakening the verification criteria, making it compatible with current high‑efficiency drafter designs.
>
> 3. **New coordination structure**
>    TriSpec also differs at the architectural level. Many methods that introduce mid‑level LLMs adopt a strictly cascaded hierarchy (e.g., TriForce), where each model interacts only with the next module. In TriSpec, the drafter, proxy verifier, and target model are all explicitly or implicitly aligned, and each interacts directly with the other two. This tight coupling enables more effective cooperation in drafting, verifying, and correcting tokens.
>
> Given these differences, we would like to kindly argue that TriSpec introduces a new perspective, a distinct optimization target, and a coordination structure that works robustly with modern small single‑layer drafters, extending the design space of verification‑side improvements.

---

> ### Author Response · Authors · 2025-11-21
> **Response by Authors to iPrA (2/4)**
>
> > **W2**: TriSpec achieves better speedup ratio at the cost of losing the theoretical lossless property of speculative decoding, which is especially important in real-world applications. ... Meanwhile, The paper should quantify quality shifts under temperature and across domains, not only average accuracy.
>
> We thank the reviewer for the insightful comment. While TriSpec trades the theoretical lossless property of standard speculative decoding for improved efficiency, we have found that TriSpec remains robust even across different tasks and under varied temperature settings.
>
> 1. **Stress Tests on general tasks**
>
>    To assess robustness in open‑ended and domain‑diverse scenarios, we extend evaluations to:
>       - **SpecBench**, which spans six categories and a wide range of fine‑grained tasks, including but not limited to question answering,  translation, writing, roleplay, and summarization, offering a comprehensive measure of open-ended generation quality.
>       - **HotpotQA**, a natural‑language, multi‑hop question answering benchmark with explicit supporting‑fact supervision, offering a distinct QA style from SpecBench.
>       - **Polymath**, a multilingual mathematical reasoning benchmark covering 18 languages, providing a stress‑test setting for multilingual prompts.
>
>    As shown in the table below, which corresponds to Table 6 in the revised paper:
>    - **SpecBench**: TriSpec achieves a **3.17×** speedup over target‑only decoding and **13%** faster than the standard speculative decoding baseline, while the overall average score decreases by only **0.2** compared to the target model.
>    - **HotpotQA**: TriSpec achieves a **3.58×** speedup over the target model and an additional **29%** speedup compared to the standard speculative decoding baseline, while maintaining essentially the same accuracy as the target.
>    - **Polymath**: TriSpec obtains a **2.95× speedup** and retains **96%** of target performance on average across 18 languages, demonstrating stability in multilingual reasoning tasks.
>
>    These results collectively indicate that TriSpec consistently provides substantial speedup while maintaining high fidelity across diverse domains.
>
> |Model|Method|SpecBench Score|SpecBench Speedup|Polymath Accuracy|Polymath Speedup|HotpotQA Accuracy|HotpotQA Speedup|
> |-|-|-|-|-|-|-|-|
> |Qwen3| Single Model (Target, 32B)|8.82|1.00x|27.8%|1.00x|93.0|1.00x|
> ||Single Model (Proxy, 1.5B)|7.33|2.72x|9.4%|3.35x|84.2|2.96x|
> ||EAGLE3|8.77|2.80x|27.20%|2.34x|92.8|2.77x|
> ||EAGLE3+TriSpec|8.62|3.17x|26.11%|3.11x|92.8|3.58x|
>
> 2. **Additional Results under Temperature=1**
>
>    Following the reviewer's suggestion, we have included experiments with sampling **temperature = 1** in the table below (**Table 2 in revised paper**). We find that:
>       - TriSpec achieves roughly **3× speedup** over target‑only decoding, and **10%–33% additional speedup** over the standard speculative decoding baseline.
>       - The maximum accuracy drop across different benchmarks is **no more than 3%**, and the average accuracy drop across five benchmarks is **no more than 2%**.
>
>    These results confirm that TriSpec maintains both efficiency and accuracy under varied temperature configurations.
>
> |Model|Method|GSM8K Acc.|GSM8K Speedup|MATH500 Acc.|MATH500 Speedup|GK23-en Acc.|GK23-en Speedup|HumanEval Acc.|HumanEval Speedup|MBPP Acc.|MBPP Speedup|Avg Acc.|Avg Speedup|
> |-|-|-|-|-|-|-|-|-|-|-|-|-|-|
> |Qwen3|Single Model (Target,32B)|98%|1.00x|85%|1.00x|76%|1.00x|87%|1.00x|76%|1.00x|84.4%|1.00x|
> ||Single Model (Proxy,1.7B)|88%|2.62x|80%|2.84x|73%|2.90x|76%|2.90x|61%|2.90x|75.6%|2.83x|
> ||HASS|96%|2.61x|86%|2.05x|75%|2.05x|88%|1.96x|74%|1.94x|83.8%|2.12x |
> ||HASS + TriSpec|95%|**3.01x**|84%|**2.51x**|77%|**2.54x**|89%|**2.31x**|72%|**2.25x**|83.4%|**2.52x**|
> ||EAGLE3|98%|2.78x|84%|2.55x|77%|2.52x|86%|2.58x|74%|2.52x|83.8%|2.59x |
> ||EAGLE3 + TriSpec|96%|**3.33x**|86%|**3.06x**|76%|**3.13x**|88%|**2.94x**|73%|**2.93x**|83.8%|**3.08x**|
> |DSQ|Single Model (Target,32B)|92%|1.00x|88%|1.00x|77%|1.00x|88%|1.00x|76%|1.00x|84.2%|1.00x|
> ||Single Model (Proxy,1.5B)|71%|2.99x|78%|2.91x|66%|3.15x|62%|3.12x|47%|3.11x|64.8%|3.06x|
> ||HASS|90%|2.97x|87%|2.19x|76%|2.33x|90%|1.98x|75%|2.04x|83.6%|2.30x|
> ||HASS + TriSpec|89%|**3.27x**|87%|**2.47x**|77%|**2.74x**|87%|**2.12x**|73%|**2.18x**|82.6%|**2.56x**|
> ||EAGLE3|92%|3.14x|85%|2.40x|77%|2.53x|89%|2.23x|76%|2.29x|83.8%|2.52x |
> ||EAGLE3 + TriSpec|88%|**3.40x**|87%|**2.79x**|79%|**2.92x**|88%|**2.36x**|72%|**2.42x**|82.8%|**2.78x**|
> |DSL|Single Model (Target,70B)|87%|1.00x|88%|1.00x|75%|1.00x|87%|1.00x|80%|1.00x|83.4%|1.00x|
> ||Single Model (Proxy,8B)|72%|3.80x|81%|3.92x|70%|3.95x|79%|3.93x|60%|3.96x|72.4%|3.91x|
> ||EAGLE3|85%|2.79x|88%|1.98x|76%|2.05x|88%|2.01x|81%|1.96x|83.2%|2.16x|
> ||EAGLE3 + TriSpec|86%|**3.93x**|89%|**2.74x**|78%|**3.06x**|89%|**2.32x**|78%|**2.39x**|84.0%|**2.89x**|

---

> ### Author Response · Authors · 2025-11-21
> **Response by Authors to iPrA (3/4)**
>
> > **W3**: Missing Related Works. Some related works [1, 2] already explored the idea of multi-level speculative decoding. The lack of these baselines weakens the novelty and evidence.
>
> We thank the reviewer for pointing out the missing baselines. While our approach differs substantially from prior multi‑level speculative decoding methods in motivation, optimization objective, and compatibility with modern high‑efficiency single‑layer drafters, we agree that including representative prior methods in our comparisons provides a more complete empirical picture.
>
> In the revised paper, we have supplemented the Related Work section and expanded **Table 1** to include representative multi‑level speculative decoding methods, ensuring a fairer comparison.
>
> For **SpecCascade**, we report results with the Chow rule, the optimal deferral rule, and the token‑specific deferral rule. Both Chow and optimal deferral rely on the drafter's output distribution, which is unreliable for single‑layer drafters like EAGLE3, leading to direct failure. Token‑specific deferral avoids this dependency and preserves accuracy, but the limited number of high‑quality draft tokens makes speedup gains small.
> For **Judge Decoding**, implementation requires re‑training a separate judge module with reconstructed data, and the relevant code has not been released as open source. Moreover, the original paper explicitly states in its limitations that "**small drafters in the spirit of Medusa or Eagle are thus not ideal**", indicating that the method is not well-suited for high‑efficiency single‑layer drafter settings.
>
> Overall, when combined with high‑efficiency single‑layer draft models, TriSpec offers a more advantageous balance between speed and accuracy.
>
> > **W4**: Evaluation is narrow and controlled. The experiments are only conducted on 2 Qwen-32B series models. The effectiveness of TriSpec on large-scale LLMs (>=70B) and other LLM backbones (e.g., llama and GLM) remains unknown.
>
> We appreciate the reviewer's concern about generality. To explicitly test whether the effectiveness of TriSpec depends on the Qwen architecture, we added experiments using DeepSeek-R1-Distill-LLaMA, which is based on the LLaMA family and therefore architecturally distinct from Qwen. We evaluate TriSpec with a 70B target model and an 8B same-family proxy verifier.
>
> As shown in the tables below (corresponding to Table 1 and Table 3 in the revised paper), the results demonstrate that:
>
> 1. A **3.36× speedup** over target-only decoding, and approximately **40 % additional speedup** over standard speculative decoding on average.
> 2. Approximately **40%-60% reduction** in target model invocations
> 3. **Less than 2%** accuracy degradation in average accuracy across all five benchmarks
>
> These findings provide concrete evidence that TriSpec scales effectively to large-scale LLMs and produces consistent speed–accuracy gains across architecturally different backbones.
>
> |Method|GSM8K Acc.|GSM8K Speedup|MATH500 Acc.|MATH500 Speedup|GK23-en Acc.|GK23-en Speedup|HumanEval Acc.|HumanEval Speedup|MBPP Acc.|MBPP Speedup|Avg Acc.|Avg Speedup |
> |-|-|-|-|-|-|-|-|-|-|-|-|-|
> |Single Model (Target,70B)|88%|1.00x|90%|1.00x|78%|1.00x|87%|1.00x|79%|1.00x|84.4%|1.00x |
> |Single Model (Proxy,8B)|77%|3.71x|83%|3.73x|73%|4.01x|80%|3.89x|59%|4.03x|74.4%|3.87x |
> |EAGLE3|88%|2.83x|89%|2.35x|76%|2.31x|84%|2.49x|80%|2.40x|83.4%|2.48x |
> |EAGLE3 + SC[Chow]|82%|2.90x|81%|2.50x|65%|2.61x|74%|2.58x|71%|2.57x|74.6%|2.63x |
> |EAGLE3 + SC[OPT]|78%|3.02x|66%|2.84x|48%|2.93x|60%|2.89x|51%|2.91x|60.6%|2.92x |
> |EAGLE3 + SC[Token]|87%|2.92x|88%|2.43x|75%|2.56x|85%|2.69x|78%|2.65x|82.6%|2.65x |
> |EAGLE3 + TriSpec|86%|**3.74x**|88%|**3.35x**|77%|**3.36x**|87%|**3.07x**|76%|**3.29x**|82.8%|**3.36x** |
>
> |Method|MATH500 $r_t$|MATH500 $t_v$(s)|MATH500 $\tau$|MATH500 $L$(s)|MATH500 Speed(Tok/s)|HumanEval $r_t$|HumanEval $t_v$(s)|HumanEval $\tau$|HumanEval $L$(s)|HumanEval Speed(Tok/s)|
> |-|-|-|-|-|-|-|-|-|-|-|
> |Single Model (Target,70B)|100.00%|/|/|256.79|8.22|100.00%|/|/|254.18|8.37 |
> |EAGLE3|28.70%|0.144|3.48|108.89|19.33|26.88%|0.147|**3.66**|107.92|20.84 |
> |EAGLE3 + TriSpec|**11.60%**|**0.093**|**3.64**|**78.32**|**27.54**|**15.18%**|**0.114**|3.59|**87.51**|**25.70** |

---

> ### Author Response · Authors · 2025-11-21
> **Response by Authors to iPrA (4/4)**
>
> > **Q1**: Could you please specify the detailed training cost of the draft model?
>
> We thank the reviewer for the question. The training pipeline for the drafter in TriSpec follows the standard **EAGLE‑series** procedure. While TriSpec introduces an additional adapter module to interface with the proxy verifier, the overall training cost remains in the same order of magnitude as standard EAGLE.
>
> To illustrate, we report the training costs for Qwen3‑32B using 8× NVIDIA A100 80 GB GPUs under different configurations:
>
> | Method           | Training strategy | Data size | Seq Len | TP (B tokens) | Epochs  | Training Time (h) |
> | ---------------- | ----------------- | --------- | ------- | ------------- | ----- | ------ |
> | EAGLE3           | —                 | 518K      | 2048    | 1.35          | 6     | 78     |
> | EAGLE3 + TriSpec | joint training    | 518K      | 2048    | 1.77          | 6     | 115    |
> | EAGLE3 + TriSpec | adapter-only      | 518K      | 2048    | 0.42          | 3     | 44     |
>
> More details about draft model training are provided in **Appendix D.1**.
>
> > **Q2**: Could you please provide more experiments on some extremely difficult tasks? Will TriSpec significantly decrease the model performance? If the user's query is out of the domain of the training data of the proxy model, may the proxy model give low-quality judge?
>
> For extremely difficult tasks, we have reported results on Polymath above. In this high‑difficulty setting, the target model itself achieves only **28%** accuracy, and the proxy’s accuracy is merely about **one‑third** of the target’s. Even under these challenging conditions, TriSpec recovers approximately 95% of the target’s accuracy while still delivering over 30% additional speedup compared to the standard speculative decoding baseline, demonstrating robustness even when the proxy is substantially weaker.
>
> For the proxy model, we directly use a smaller same‑family model without additional training. Because such proxy models share highly similar pre‑training procedures and data with their corresponding targets, they are typically well‑aligned in distribution. This alignment greatly reduces the risk of OOD issues and ensures reliable proxy judgments.
> For example, models in the Qwen3 family are trained on approximately 36T tokens of diverse text, spanning multiple domains and languages. Under the same training paradigm and data coverage, the smaller same‑family models inherit much of the target’s broad generalization ability, enabling them to provide high‑quality verification even when user queries fall outside the proxy’s primary training domain.
>
> >**Q3**: Does the proxy and the target model run on the same GPU? Whether the KV cache is shared?
>
> To avoid any impact on the proxy model’s performance, the proxy is kept fully independent and maintains its own KV cache, which is not shared with the target model. As the proxy is much smaller than the target, it incurs only a modest additional memory overhead (approximately **7%** in the Qwen3 experimental setting) during inference. When GPU memory is sufficient, both the proxy and the target model can be deployed on the same GPU. Since the two models operate independently, they can also be placed on different GPUs to accommodate hardware constraints or optimize resource allocation.

---

> > ### Comment · Reviewer_iPrA · 2025-11-25
> >
> > Thank you for the detailed response and additional experiments. However, some of my concerns still remain.
> >
> > **Regarding the novelty clarification.**
> > I recognize the validity of the authors' first and third points. However, the second point remains unconvincing and appears to be a logical obstacle. Could the authors please provide further elaboration or concrete examples to clarify this specific aspect?
> >
> > **Regarding the accuracy degradation.**
> > I have several follow-up concerns regarding the additional experiments:
> >
> > 1.  **Evaluation Implementation:** How was the model performance evaluated on Spec-Bench? To my knowledge, the official codebase does not provide a standardized evaluation harness for this purpose.
> > 2.  **Discrepancy in Results:** Why is there a significant discrepancy between your reported accuracy metrics and the official results? For instance, the authors report an accuracy of **85%** with Qwen3-32B on **MATH-500**, whereas the official report claims **97.1%**. I have observed similar inconsistencies across other benchmarks as well.
> > 3.  **Benchmark Selection:** For evaluating extremely challenging tasks, **AIME 24 \& 25** are widely considered more standard and representative benchmarks than PolyMath, and **GPQA-diamond** is preferred over HotpotQA.
> > 4.  **Baseline Model:** I concur with Reviewer N6ke that using a base **Llama** family model would be a more appropriate baseline than `DeepSeek-R1-Distill-LLaMA` for a fair comparison.
> >
> > **Regarding the additional memory and system complexity.**
> > I believe that deploying three models simultaneously presents greater practical difficulties than implied. Both the memory consumption for high-concurrency/long-context KV cache and the structural overhead to the inference pipeline pose significant challenges.
> >
> > * **Memory Footprint:** As highlighted in **MagicDec [1]**, the memory footprint of the draft model scales rapidly with increased batch size and context length, which will be more severe in the proposed three-model architecture.
> > * **Load Balancing & Overhead:** A three-model deployment risks severe resource contention and load imbalance. If the draft and proxy models are deployed with **TP=1**, the memory on GPU 0 is prone to overflowing (OOM). Conversely, if deployed with **TP=8**, the additional communication overhead will significantly degrade the inference speed of both the draft and proxy models.
> >
> > ---
> >
> > [1] Sadhukhan, Ranajoy, et al. "Magicdec: Breaking the latency-throughput tradeoff for long context generation with speculative decoding." arXiv preprint arXiv:2408.11049 (2024).

---

> > > ### Author Response · Authors · 2025-11-26
> > > **Clarification on Novelty**
> > >
> > > We sincerely thank the reviewer for the detailed feedback and for identifying multiple points that require further clarification.
> > >
> > > > Regarding the novelty clarification
> > >
> > > Here we explain in detail the limitations of prior verification‑side acceleration methods, and why TriSpec does not suffer from these constraints.
> > >
> > > **Differences in draft model choice**
> > > Prior works such as **Judge Decoding** and **SpecCascade** accelerate standard speculative decoding by **relaxing verification**, i.e., allowing certain draft tokens to bypass strict target‑model verification and be accepted into the final output. In their published experiments, the small draft models are all **full multi‑layer architectures** with independent reasoning capability — e.g., LLaMA‑3.1‑8B in Judge Decoding, T5‑Small and Gemma‑2B in SpecCascade. Such drafters can generate high‑quality tokens autonomously, so relaxed verification can accept more draft outputs without noticeably harming accuracy, yielding worthwhile speed gains.
> > >
> > > However, in current high‑efficiency speculative decoding, such multi‑layer small models are no longer the preferred choice because their deep backbone makes drafting slow. For example, in **Fig. 2 (Sps)** of our paper, using Qwen3‑1.7B as a drafter with Qwen3‑32B target takes $t_d = 0.162$ s to generate 5 draft tokens — more than 10× slower than a single‑layer Eagle3 drafter ($t_d = 0.015$ s). For this reason, the **de‑facto standard today** is single‑layer designs such as **EAGLE3** and **MTP**, which retain the target model’s token embeddings and LM head, compress the backbone to one Transformer layer, and are trained to align closely with the target model’s outputs by taking its hidden states as additional inputs for guidance.
> > >
> > > **Limitations of prior methods under single‑layer drafters**
> > > While extremely fast, these single‑layer drafters have **weak standalone reasoning** (**Appendix C, Fig. 7**), and their token probability distributions are often unreliable compared to the target’s (**Appendix C, Fig. 8**).
> > > Relaxed‑verification methods depend heavily on the quality of the draft tokens and their probabilities. As a result, when paired with single‑layer drafters:
> > > - **Judge Decoding** explicitly lists “The draft model needs to be of high-quality, otherwise our approach naturally does not prove beneficial and too many tokens end up being rejected. Self-speculation and small drafters in the spirit of Medusa or Eagle are thus not ideal since their generations quickly deteriorate when drafting too far into the future.” in its limitations.
> > > - **Table 1** in our paper shows that **SpecCascade**, when applied to single‑layer drafters, either suffers a significant accuracy drop (≥ 10%) due to unreliable draft probability estimates, or, if accuracy is preserved, delivers only limited speedup (≤ 5%).
> > >
> > > The fundamental reason these methods fail to deliver strong gains here is that the number of tokens eligible for relaxed acceptance becomes very small, due to the quality degradation in drafts when drafters operate beyond a few steps.
> > >
> > > **Why TriSpec avoids this limitation**
> > > In contrast, TriSpec’s acceleration does not rely on either the intrinsic quality of the draft tokens or the accuracy of their probability distributions.
> > > Instead of weakening verification criteria, it reduces the per‑round verification time $t_v$ by introducing a lightweight proxy verifier aligned with the target.
> > > This allows TriSpec to work effectively with single‑layer high‑efficiency drafters, maintaining speed gains and accuracy in exactly the scenario where many prior verification‑side methods lose their advantage.

---

> > > ### Author Response · Authors · 2025-11-26
> > > **Clarification on Evaluation and Benchmarks**
> > >
> > > > Regarding the accuracy degradation
> > >
> > > **Evaluation implementation on Spec‑Bench**
> > > For **Spec‑Bench**, we adopted an **LLM-as-a-judge** evaluation strategy. Specifically, Spec‑Bench is an extension of MT‑Bench, and we followed the MT‑Bench evaluation pipeline ([FastChat, llm_judge](https://github.com/lm-sys/FastChat/tree/main/fastchat/llm_judge)) as reference. GPT‑4.1 was used as the judge model to score each response on a 1–10 scale based on correctness and completeness, and scores were then averaged at the turn level for system-level comparison. Details of these additional benchmarks and the scoring procedure are provided in Section **B.1** of the revised paper.
> > >
> > > **Discrepancy between our reported accuracy and official results**
> > > To make our testing more representative for speculative decoding, we ran all experiments with **reasoning models** or enabled “thinking” mode. This setup produced outputs of sufficient length to avoid bias in speed measurements from very short generations. In the main results, all experiments used a single NVIDIA A100 80G GPU and set `max_new_tokens = 8192` to prevent out-of-memory issues, which caused tasks requiring longer outputs to be truncated and marked as incorrect. This length constraint is the main reason for the discrepancy between our reported accuracy and the official results. Since the same constraint was applied to all compared methods, the relative accuracy comparisons remain fair. For completeness, we also conducted additional experiments with `4 GPUs` and `max_new_tokens = 32768` to allow full outputs, covering MATH‑500, HumanEval, AIME‑24, AIME‑25, and GPQA‑diamond. Results confirm that when outputs are not truncated, TriSpec maintains minimal accuracy loss relative to the target model.
> > >
> > > |Model|Method|MATH500 Accuracy|MATH500 Speedup|HumanEval Accuracy|HumanEval Speedup|AIME24 Accuracy|AIME24 Speedup|AIME25 Accuracy|AIME25 Speedup|GPQA-diamond Accuracy|GPQA-diamond Speedup|
> > > |-|-|-|-|-|-|-|-|-|-|-|-|
> > > |Qwen3|Single Model (Target, 32B)|95%|1.00x|90%|1.00x|24/30|1.00x|22/30|1.00x|67%|1.00x|
> > > | |Single Model (Proxy, 1.7B)|90%|3.22x|83%|3.28x|16/30|3.60x|12/30|3.54x|45%|3.33x|
> > > | |EAGLE3|95%|2.81x|91%|2.93x|22/30|2.52x|21/30|2.31x|68%|2.97x|
> > > | |EAGLE3+TriSpec|95%|3.71x|90%|3.68x|23/30|3.37x|21/30|3.30x|66%|4.39x|
> > >
> > > **Benchmark selection**
> > > We acknowledge the reviewer’s suggestion to adopt more widely‑used and representative benchmarks, and accordingly added evaluations on AIME‑24, AIME‑25, and GPQA‑diamond. On these benchmarks, TriSpec achieves less than 3% accuracy degradation compared to standard speculative decoding while delivering 20–40% speedup. Moreover, evaluations on PolyMath and HotpotQA remain valuable, as they demonstrate TriSpec’s strong performance on multilingual and multi‑hop QA tasks, underscoring its robustness across diverse task types.
> > >
> > > **Baseline model choice**
> > > Regarding the choice of baseline model, we initially selected DeepSeek‑R1‑Distill‑LLaMA as the representative of the LLaMA architecture. This selection was motivated by its inheritance of the reasoning patterns of DeepSeek‑R1, which yield sufficiently long chain‑of‑thought outputs and mitigate the risk of short‑answer bias in speculative decoding evaluation. The results obtained with this model already indicate that the advantages of TriSpec are not tied to a specific architecture. Following the reviewers’ suggestions, we further included experiments using **LLaMA‑3.1‑8B** and **LLaMA‑3.2‑1B** as both the target model and proxy verifier. As shown in table below, TriSpec remains effective on these original LLaMA models, maintaining accuracy close to the target while improving speed compared to standard speculative decoding.
> > >
> > > |Model|Method|GSM8K Accuracy|GSM8K Speedup|MATH500 Accuracy|MATH500 Speedup|Gk23-en Accuracy|Gk23-en Speedup|humaneval Accuracy|humaneval Speedup|MBPP Accuracy|MBPP Speedup|
> > > |-|-|-|-|-|-|-|-|-|-|-|-|
> > > |LLAMA|Single Model (Target, 8B)|64%|1.00x|34%|1.00x|18%|1.00x|66%|1.00x|57%|1.00x|
> > > | |Single Model (Proxy, 1B)|34%|2.54x|26%|2.53x|10%|2.50x|46%|2.58x|29%|2.53x|
> > > | |HASS|64%|2.93x|34%|3.27x|18%|3.23x|66%|3.52x|57%|3.46x|
> > > | |HASS + TriSpec|62%|3.17x|34%|3.73x|17%|3.66x|66%|3.81x|54%|3.76x|

---

> ### Author Response · Authors · 2025-11-26
> **Clarification on Deployment and Complexity**
>
> > Regarding the additional memory and system complexity.
>
> In terms of **memory footprint**, although the memory usage of the draft and proxy models increases with batch size and context length, their requirements remain a small fraction of that of the target model. The table below presents measurements from our Qwen3‑series experiments, showing the GPU memory consumption of the three models at different context lengths. The combined memory footprint of the proxy and draft models accounts for no more than **12%** of the total memory consumption at a context length of 16k, and remains below **20%** even at 128k, where the benefits of speculative decoding have already decreased substantially. Therefore, the additional cost of introducing a proxy verifier is modest relative to the dominant memory demand of the target model and is practically acceptable.
>
> | Context length | Target Model (GB) | Proxy Model (GB) | Draft Model (GB) |
> |-|-|-|-|
> | 8k  | 67.47 | 4.91 | 4.18 |
> | 16k | 69.42 | 5.76 | 4.21 |
> | 32k | 73.32 | 7.46 | 4.28 |
> | 64k | 81.12 | 10.86| 4.41 |
> | 128k| 96.72 | 17.66| 4.67 |
>
>
> Regarding **load balancing and communication overhead**, we note that in current production‑level deployments of standard speculative decoding, for instance in **[SGLang’s EAGLE3 pipeline](https://docs.sglang.io/advanced_features/speculative_decoding.html)**, the draft and target models are deployed with the same TP size to balance memory usage and computational throughput. While a higher TP size does introduce more inter‑device communication, its cost is small for the draft model in practice because the single‑layer architecture drastically reduces the number of all-reduce synchronizations per forward pass. Increasing the TP size in this setting not only mitigates potential OOM issues by distributing memory demand but also accelerates parallel computation. As a result, the draft stage typically takes only about 10% of the verification stage’s runtime per decoding round in such deployments.
>
> The same rationale applies to the proxy verifier in TriSpec. By using the same TP size as the target, we preserve load balancing, while benefiting from the proxy’s much lower per‑forward computation and communication costs due to its considerably smaller parameter count. For example, in our measurements with **TP=8** on SGLang, the time for a single forward pass of Qwen3‑1.7B is only about **30%** of that of Qwen3‑32B (see table below); When the number of Case‑1 (handled only by the proxy) exceeds that of Case‑2 (requiring fall‑back to the target), which is the predominant situation in most of our experiments, the acceleration from proxy‑based verification is consistently significant, yielding a substantial reduction in per‑round verification time $t_v$ and thus an overall speedup.
>
> | Context Length | Qwen3-1.7B  single forward time (ms) | Qwen3-32B single forward time (ms) |
> | -------------- | ---------- | --------- |
> | 2k             | 3.05       | 9.70      |
> | 8k             | 3.08       | 9.81      |
> | 16k            | 3.12       | 10.22     |
> | 32k            | 3.20       | 10.86     |
> | 64k            | 3.34       | 11.98     |
> | 128k           | 3.71       | 14.78     |

---

### Official Review · Reviewer_N6ke · 2025-10-29

**Soundness:** 2
**Presentation:** 3
**Contribution:** 3
**Rating:** 6
**Confidence:** 4

**Summary:**

TriSpec is a speculative decoding framework that uses a small model of the same family as an approximate proxy of the target model for use in verification. Unlike classic speculative decoding, not every token is verified by the target model. Drafted tokens are first verified by the proxy model. Only when the proxy is unable to make a confident verification (indicated by low margin between top-1 and top-2 token probability) is the target model used for verification.

On math and code reasoning benchmarks, the authors show that TriSpec achieves larger speedups than baseline speculative decoding methods while seeing negligible performance drop despite the target model never validating the full output.

**Strengths:**

- TriSpec is a simple and effective idea, using small models as verifiers for a fast single-layer drafter, similar to model cascades but for verification.
- Across all domains presented in the paper, TriSpec demonstrates higher speedups compared to baselines while showing negligible performance loss compared to the target model. These results show that with the right proxy, the loss of the losslessness guarantee from classical speculative decoding will not adversely affect output quality.

**Weaknesses:**

- The paper only examines two model families, both based on Qwen: Qwen3 and DeepSeek-R1-Distilled-Qwen. Experiments on model families from other providers would strengthen the paper. In the paper’s current state, it is unclear whether the effectiveness of smaller model variants as proxy verifiers is particular to Qwen as a model provider.
- The paper only examines two settings: math and code reasoning. These settings may be much more structured than more general domains, better suiting proxy models. Evaluations on other domains like question answering (e.g., HotpotQA) or instruction following would make the paper stronger. It could also be interesting to see results in domains where there is a much larger performance gap between the proxy and target models.
- The evaluation set sizes are small, only 100 questions per benchmark. This, along with the lack of error bars and confidence intervals in the paper, makes it difficult to fully contextualize the results.

**Questions:**

- Did you investigate the impact of a mis-aligned proxy (e.g., from a different model family) on accuracy/latency?
- How did you select the proxy model size? In particular, why not Qwen3 0.6B?
- Have you experimented with more layers of proxies between the draft and target model, and if so, why did you decide to have only one proxy?

---

> ### Author Response · Authors · 2025-11-21
> **Response by Authors to N6ke (1/3)**
>
> > **W1**: The paper only examines two model families, both based on Qwen: Qwen3 and DeepSeek-R1-Distilled-Qwen. Experiments on model families from other providers would strengthen the paper. In the paper’s current state, it is unclear whether the effectiveness of smaller model variants as proxy verifiers is particular to Qwen as a model provider.
>
> We appreciate the reviewer’s concern about generality. To explicitly test whether the effectiveness of TriSpec depends on the Qwen architecture, we added experiments using DeepSeek-R1-Distill-LLaMA, which is based on the LLaMA family and therefore architecturally distinct from Qwen. We evaluate TriSpec with a 70B target model and an 8B same-family proxy verifier.
>
> As shown in the tables below (corresponding to Table 1 and Table 3 in the revised paper), the results demonstrate that:
>
> 1. A **3.36× speedup** over target-only decoding, and approximately **40 % additional speedup** over standard speculative decoding on average.
> 2. Approximately **50% reduction** in target model invocations
> 3. **Less than 2%** accuracy degradation in average accuracy across all five benchmarks
>
> These results provide concrete evidence that TriSpec’s proxy-verifier mechanism is **architecture-agnostic** and that smaller model variants can serve as effective verifiers beyond a single model provider.
>
> | Method|GSM8K Acc.|GSM8K Speedup|MATH500 Acc.|MATH500 Speedup|GK23-en Acc.|GK23-en Speedup|HumanEval Acc.|HumanEval Speedup|MBPP Acc.|MBPP Speedup|Avg Acc.|Avg Speedup |
> |-|-|-|-|-|-|-|-|-|-|-|-|-|
> | Single Model (Target,70B)|88%|1.00x|90%|1.00x|78%|1.00x|87%|1.00x|79%|1.00x|84.4%|1.00x |
> | Single Model (Proxy,8B)|77%|3.71x|83%|3.73x|73%|4.01x|80%|3.89x|59%|4.03x|74.4%|3.87x |
> | EAGLE3|88%|2.83x|89%|2.35x|76%|2.31x|84%|2.49x|80%|2.40x|83.4%|2.48x |
> | EAGLE3 + SC[Chow]|82%|2.90x|81%|2.50x|65%|2.61x|74%|2.58x|71%|2.57x|74.6%|2.63x |
> | EAGLE3 + SC[OPT]|78%|3.02x|66%|2.84x|48%|2.93x|60%|2.89x|51%|2.91x|60.6%|2.92x |
> | EAGLE3 + SC[Token]|87%|2.92x|88%|2.43x|75%|2.56x|85%|2.69x|78%|2.65x|82.6%|2.65x |
> | EAGLE3 + TriSpec|86%|**3.74x**|88%|**3.35x**|77%|**3.36x**|87%|**3.07x**|76%|**3.29x**|82.8%|**3.36x** |
>
> | Method|MATH500 $r_t$|MATH500 $t_v$(s)|MATH500 $\tau$|MATH500 $L$(s)|MATH500 Speed(Tok/s)|HumanEval $r_t$|HumanEval $t_v$(s)|HumanEval $\tau$|HumanEval $L$(s)|HumanEval Speed(Tok/s) |
> |-|-|-|-|-|-|-|-|-|-|-|
> | Single Model (Target,70B)|100.00%|/|/|256.79|8.22|100.00%|/|/|254.18|8.37 |
> | EAGLE3|28.70%|0.144|3.48|108.89|19.33|26.88%|0.147|**3.66**|107.92|20.84 |
> | EAGLE3 + TriSpec|**11.60%**|**0.093**|**3.64**|**78.32**|**27.54**|**15.18%**|**0.114**|3.59|**87.51**|**25.70** |

---

> ### Author Response · Authors · 2025-11-21
> **Response by Authors to N6ke (2/3)**
>
> > **W2**: The paper only examines two settings: math and code reasoning. These settings may be much more structured than more general domains, better suiting proxy models. Evaluations on other domains like question answering (e.g., HotpotQA) or instruction following would make the paper stronger. It could also be interesting to see results in domains where there is a much larger performance gap between the proxy and target models.
>
> Thank you for the suggestion. We address this concern in two ways:
> 1. **Beyond math/code (more general domains)**
>    We extend our evaluation to:
>    - **SpecBench**, which spans six categories and a wide range of fine‑grained tasks, including but not limited to question answering,  translation, writing, roleplay, and summarization, offering a comprehensive measure of open-ended generation quality.
>    - **HotpotQA**, a natural-language, multi-hop question answering benchmark with explicit supporting-fact supervision, offering a distinct style of QA evaluation compared to the tasks in SpecBench.
>
>    As shown in the table below, which corresponds to Table 6 in the revised paper:
>    - **SpecBench**: TriSpec achieves a **3.17×** speedup over target‑only decoding and **13%** faster than the standard speculative decoding baseline, while the overall average score decreases by only **0.2** compared to the target model.
>    - **HotpotQA**: TriSpec achieves a **3.58×** speedup over the target model and an additional **29%** speedup compared to the standard speculative decoding baseline, while maintaining essentially the same accuracy as the target.
>
>    In addition, **Figure 5** in the revised paper presents TriSpec's performance across the **individual task categories** within SpecBench, demonstrating its effectiveness across all task types.
>
> 2. **Domains with a large proxy–target performance gap**
>    To study the high-gap regime, we evaluate **Polymath** at the high difficulty level, where the proxy model's accuracy is roughly **one-third** that of the target.
>    Even in this challenging setting, TriSpec recovers approximately **95%** of the target's accuracy while still delivering over **30% additional speedup compared to the standard speculative decoding baseline**, demonstrating robustness even when the proxy is substantially weaker.
>
> These results together show that TriSpec is effective not only on structured math/code reasoning, but also on open-ended, general-purpose tasks and in difficult regimes with large proxy–target gaps.
>
> |Model|Method|SpecBench Score|SpecBench Speedup|Polymath Accuracy|Polymath Speedup|HotpotQA Accuracy|HotpotQA Speedup|
> |-|-|-|-|-|-|-|-|
> |Qwen3| Single Model (Target, 32B)|8.82|1.00x|27.8%|1.00x|93.0|1.00x |
> |      |Single Model (Proxy, 1.5B)|7.33|2.72x|9.4%|3.35x|84.2|2.96x |
> | |EAGLE3|8.77|2.80x|27.20%|2.34x|92.8|2.77x |
> |      |EAGLE3+TriSpec|8.62|3.17x|26.11%|3.11x|92.8|3.58x |
>
> > **W3**: The evaluation set sizes are small, only 100 questions per benchmark. This, along with the lack of error bars and confidence intervals in the paper, makes it difficult to fully contextualize the results.
>
> We acknowledge that using smaller test sets can introduce statistical variance. in the table below (**Table 7** in revised paper), we additionally report results on the full test sets for all five benchmarks. These full-set evaluations confirm that TriSpec consistently maintains the same accuracy–speedup trade-off observed in the 100-sample subsets presented in the main paper.
>
> Additionally, we perform repeated evaluation on the GSM8K benchmark by sampling eight distinct subsets of 100 questions, and plot the mean results with error bars denoting one standard deviation across runs (**Figure 6** in revised paper). The narrow error bars indicate that performance variations introduced by random subset sampling are minor.
>
> Together, these expanded experiments show that TriSpec's performance trends hold under both larger test sets and repeated‑subset sampling, demonstrating the robustness of our conclusions.
>
> | Method|GSM8K Acc.|GSM8K Speedup|MATH500 Acc.|MATH500 Speedup|GK23-en Acc.|GK23-en Speedup|Humaneval Acc.|Humaneval Speedup|MBPP Acc.|MBPP Speedup |
> |-|-|-|-|-|-|-|-|-|-|-|
> | Single Model (Target, 32B)|94.9%|1.00x|86.8%|1.00x|78.2%|1.00x|86.6%|1.00x|74.8%|1.00x |
> | Single Model (Proxy, 1.5B)|86.0%|2.52x|78.2%|2.87x|69.9%|2.76x|77.4%|2.87x|52.8%|2.93x |
> | EAGLE3|94.3%|3.29x|88.0%|2.77x|77.1%|2.77x|87.2%|2.72x|74.2%|2.71x |
> | EAGLE3 + TriSpec|92.9%|4.08x|87.4%|3.48x|77.4%|3.39x|86.0%|3.23x|71.4%|3.33x |

---

> ### Author Response · Authors · 2025-11-21
> **Response by Authors to N6ke (3/3)**
>
> > **Q1**: Did you investigate the impact of a mis-aligned proxy (e.g., from a different model family) on accuracy/latency?
>
> We have not adopted proxies from a completely different model family in TriSpec due to fundamental compatibility constraints. In speculative decoding, cross-family collaboration is rarely used because models from different families typically have **incompatible vocabularies**. This prevents the verifier from reliably performing token-level evaluation of draft outputs or providing accurate local corrections.
>
> In addition, draft models such as EAGLE are specifically trained to imitate the target model's behavior at the token level. In TriSpec, the proxy's role is to replace the target model in verifying these drafted tokens. If the proxy itself differs substantially from the target in generation behavior, it will tend to pull the verification decisions toward its own output distribution. This misalignment can reduce the acceptance rate and undermine the cooperative dynamics between the draft and target model.
>
> For these reasons, TriSpec focuses on **same-family proxies**, which share the tokenizer and closely match the output distribution of the target, enabling consistent token-level verification and stable accuracy–latency trade-offs. **We include a special discussion regarding this point in Appendix C.4 of the revised paper.**
>
> > **Q2**: How did you select the proxy model size? In particular, why not Qwen3 0.6B?
>
> When selecting the proxy model, we considered two key factors:
>
> 1. **Alignment with the target model**: The proxy must have noticeably stronger alignment with the target than the drafter; otherwise, it cannot provide reliable local corrections to the drafter's output, and thus cannot effectively reduce target model invocations.
>
> 2. **Verification cost**: While the proxy needs sufficient capacity to achieve the required alignment, its size must remain small enough to avoid prohibitive verification cost.
>
> As shown in Figure 10 of the revised paper, we plot model parameter count against alignment (exact-match ratio) within the Qwen3 family. The 1.7B proxy shows a clear alignment improvement over the 0.6B model. In contrast, the 4B model improves alignment only slightly beyond 1.7B but brings a higher inference cost. Based on this trade-off, we selected **1.7B** as the proxy configuration for TriSpec.
>
> > **Q3**: Have you experimented with more layers of proxies between the draft and target model, and if so, why did you decide to have only one proxy?
>
> We appreciate the reviewer's suggestion. In TriSpec, we have not adopted multiple proxy layers. As shown in Figure 10, when using a 32B target model with an existing 1.7B proxy, adding another mid-sized proxy layer yields only a slight improvement in alignment. Such marginal gain does not justify the additional memory and computation cost, making extra layers a risky choice under these conditions.
>
> Based on this observation, we believe that a single-proxy design offers the best balance of efficiency, accuracy, and system simplicity for target models of a small scale. That said, for scenarios where the target model is substantially larger and its alignment gap with the proxy remains significant, incorporating an additional proxy layer could be an interesting direction for future exploration.

---

> > ### Comment · Reviewer_N6ke · 2025-11-24
> >
> > Thank you for the detailed response, most of my concerns are well addressed.
> >
> > I do have one key concern about the response: why was DeepSeek-R1-Distill-LLaMA and not a base Llama model family chosen as the representative of the Llama model architecture? My understanding is that all sizes of the DeepSeek-R1-Distill-LLaMA models are trained to match the output tokens of DeepSeek-R1, making such models very much suited for the proxy verification task. I had similar concerns about DeepSeek-R1-Distill-LLaMA, but they were alleviated by the fact that the original Qwen model family was also evaluated.

---

> > > ### Author Response · Authors · 2025-11-26
> > >
> > > We thank the reviewer for the constructive comments and suggestions. We selected DeepSeek‑R1‑Distill‑LLaMA as the representative of the LLaMA architecture because it inherits the reasoning patterns of DeepSeek‑R1, and its chain‑of‑thought style leads to responses that are typically long enough to avoid weakening the acceleration benefit or introducing bias into the evaluation. We agree with the reviewer’s point that models in the DeepSeek‑R1‑Distill‑LLaMA series are well suited for the proxy verification role, as they are trained to match the output style and patterns of DeepSeek‑R1. In fact, all models within the same family exhibit this alignment through highly similar pre‑training and post‑training data as well as training paradigms, producing output styles that closely resemble one another. This process is essentially equivalent in effect to model distillation. Following the reviewer’s suggestion, we further conducted experiments using LLaMA‑3.1‑8B and LLaMA‑3.2‑1B as both the target model and proxy verifier. The results in the table below show that TriSpec remains effective on these original LLaMA models, maintaining accuracy close to that of the target while providing faster verification compared to standard speculative decoding.
> > >
> > > |Model|Method|GSM8K Accuracy|GSM8K Speedup|MATH500 Accuracy|MATH500 Speedup|Gk23-en Accuracy|Gk23-en Speedup|humaneval Accuracy|humaneval Speedup|MBPP Accuracy|MBPP Speedup|
> > > |-|-|-|-|-|-|-|-|-|-|-|-|
> > > |LLaMA3|Single Model (Target, 8B)|64%|1.00x|34%|1.00x|18%|1.00x|66%|1.00x|57%|1.00x|
> > > | |Single Model (Proxy, 1B)|34%|2.54x|26%|2.53x|10%|2.50x|46%|2.58x|29%|2.53x|
> > > | |HASS|64%|2.93x|34%|3.27x|18%|3.23x|66%|3.52x|57%|3.46x|
> > > | |HASS + TriSpec|62%|3.17x|34%|3.73x|17%|3.66x|66%|3.81x|54%|3.76x|

---

### Official Review · Reviewer_KuyU · 2025-11-03

**Soundness:** 3
**Presentation:** 3
**Contribution:** 3
**Rating:** 4
**Confidence:** 4

**Summary:**

This paper presents TriSpec, a novel speculative decoding (SD) framework that introduces a proxy verifier to reduce verification cost—an often-overlooked bottleneck in SD pipelines. Unlike previous work (e.g., Medusa, EAGLE, SpecDec++) that primarily optimized the drafting phase, TriSpec focuses on verification efficiency by employing a lightweight, same-family small model to pre-verify tokens before escalating uncertain ones to the full target model. A margin-based routing criterion determines when to trust the proxy versus when to defer to the target.

**Strengths:**

The writing is very clear and easy to follow. I particularly appreciate that the authors clearly illustrate the bottlenecks that current speculative decoding systems suffer from, as shown in Figure 2. The proposed approach—based on introducing a lightweight proxy verifier to reduce verification cost—is both reasonable and well motivated. In terms of experiments, the authors conduct comprehensive evaluations on five benchmarks across two metrics (accuracy and speedup), demonstrating consistent improvements and strong empirical support for the proposed framework.

**Weaknesses:**

The hierarchical framework seems not entirely new, previous work such as triforce [1] also employs similar hierarchical framework. I understand there are some difference, but the authors should give some discussion between them.  In addition, I find the preliminary observation in Figure 2(b) particularly interesting. However, I wonder whether this phenomenon persists under varied temperature settings. Intuitively, when the temperature is higher, the output distribution becomes smoother, which might weaken the reliability of the margin-based routing criterion. In such cases, the proposed approach may not perform as well. I hope authors can clarify this. Moreover, the main experiments are conducted only under a fixed temperature = 0 setting. I recommend the authors evaluate their approach under more diverse temperature conditions to better assess its robustness.

[1] Triforce: Lossless acceleration of long sequence generation with hierarchical speculative decoding

**Questions:**

See weakness.

---

> ### Author Response · Authors · 2025-11-21
> **Response by Authors to KuyU (1/2)**
>
> > **W1**: The hierarchical framework seems not entirely new, previous work such as triforce also employs similar hierarchical framework. I understand there are some difference, but the authors should give some discussion between them.
>
> Thank you for the suggestion. We would like to clarify that while Triforce and other prior work adopt hierarchical structures, TriSpec differs from them in **setting**, **motivation**, and **architecture**.
>
> 1. **Different experimental setting**:
>    TriSpec is designed as a general enhancement to the standard speculative decoding framework and is not confined to specialized settings. In contrast, Triforce targets long‑context inference where KV‑cache bottlenecks are prominent; however, for standard‑length inputs, its reliance on a high‑parameter auxiliary model can substantially increase inference cost. As shown in its main result (Figure 5), for inputs shorter than 32K tokens Triforce yields lower speedup than the naive policy. By comparison, TriSpec achieves speedup by leveraging the strong alignment between a same‑family proxy verifier and the target model. This alignment, resulting from similarly broad pretraining, is consistently maintained across diverse input contexts.
>
> 2. **Different motivation and optimization goal**:
>    TriSpec is designed to address the high verification overhead in speculative decoding, a problem rarely targeted in prior work. Triforce is motivated by the observation that a standard drafter often lacks sufficient capability in long‑context tasks; it therefore employs a sparse‑KV‑cache version of the target model as the drafter, with the primary intention of improving the drafter's capability. Other approaches, such as Judge Decoding [1] and SpecCascade [2], attempt to relax the verification constraints to boost the overall acceptance rate. None of these methods explicitly optimize the cost of the verification process itself.
>    In contrast, TriSpec aims to have part of the target model's verification performed by a proxy verifier in suitable cases, thereby reducing per‑round verification cost while maintaining accuracy. This objective is complementary to, rather than conflicting with, the goals of the aforementioned methods.
>
> 3. **Different architecture**:
>    TriSpec's design also differs at the interaction level. Triforce adopts a cascaded architecture, where each model serves only the next downstream module. In TriSpec, the drafter, proxy verifier, and target model are all explicitly or implicitly aligned, and each interacts directly with the other two, allowing tighter coupling and more effective cooperation.
>
> These distinctions clarify that, despite superficial similarities to prior hierarchical frameworks, TriSpec differs fundamentally in its usage scenarios, objectives, and interaction patterns. We appreciate the reviewer's suggestion, and **to provide a complete comparison, the revised paper includes a discussion of related hierarchical methods like TriForce in the Related Work section.**
>
> [1] Bachmann, Gregor, Sotiris Anagnostidis, Albert Pumarola, Markos Georgopoulos, Artsiom Sanakoyeu, Yuming Du, Edgar Schönfeld, Ali Thabet, and Jonas Kohler. "Judge decoding: Faster speculative sampling requires going beyond model alignment." arXiv preprint arXiv:2501.19309 (2025).
>
> [2] Narasimhan, Harikrishna, Wittawat Jitkrittum, Ankit Singh Rawat, Seungyeon Kim, Neha Gupta, Aditya Krishna Menon, and Sanjiv Kumar. "Faster cascades via speculative decoding." arXiv preprint arXiv:2405.19261 (2024).

---

> ### Author Response · Authors · 2025-11-21
> **Response by Authors to KuyU (2/2)**
>
> > **W2**: In addition, I find the preliminary observation in Figure 2(b) particularly interesting. However, I wonder whether this phenomenon persists under varied temperature settings. Intuitively, when the temperature is higher, the output distribution becomes smoother, which might weaken the reliability of the margin-based routing criterion. In such cases, the proposed approach may not perform as well. I hope authors can clarify this. Moreover, the main experiments are conducted only under a fixed temperature = 0 setting. I recommend the authors evaluate their approach under more diverse temperature conditions to better assess its robustness.
>
> Thank you for the insightful question.
> 1. **Clarification on margin computation and sampling temperature**
>    In our design, the margin is computed from the standard softmax distribution (temperature = 1) independently of the sampling temperature used for generation. This is intentional: the margin measures whether the proxy's predictions have sufficient separation, and this metric does not need to be linked to the sampling temperature. Computing the margin at a fixed temperature keeps the threshold $\lambda$ stable, eliminating the need to retune it when the sampling temperature changes. Similar designs appear in prior work such as SpecCascade, where the deferral rule between small and large model is likewise independent of sampling temperature. Consequently, variations sampling temperatures do not weaken the reliability of the margin‑based decision rule in TriSpec.
>
> 2. **Additional results under other temperature settings**
>    Following the reviewer's suggestion, we have included experiments with sampling temperature = 1, as shown in the table below (Table2 in revised paper). Under this setting, we observe that:
>    - TriSpec achieves roughly **3× speedup** over target‑only decoding, and **10%–33% additional speedup** over the standard speculative decoding baseline.
>    - The maximum accuracy drop across different benchmarks is **no more than 3%**, and the average accuracy drop across five benchmarks is **no more than 2%**.
>
>    These results confirm that TriSpec maintains both efficiency and accuracy under varied temperature configurations.
>
> | Model | Method | GSM8K Acc. | GSM8K Speedup | MATH500 Acc. | MATH500 Speedup | GK23-en Acc. | GK23-en Speedup | HumanEval Acc. | HumanEval Speedup | MBPP Acc. | MBPP Speedup | Avg Acc. | Avg Speedup |
> |-------|--------|------------|---------------|--------------|-----------------|--------------|-----------------|----------------|-------------------|-----------|--------------|----------|-------------|
> |  Qwen3| Single Model (Target,32B) | 98% | 1.00x | 85% | 1.00x | 76% | 1.00x | 87% | 1.00x | 76% | 1.00x | 84.4% | 1.00x |
> |       | Single Model (Proxy,1.7B) | 88% | 2.62x | 80% | 2.84x | 73% | 2.90x | 76% | 2.90x | 61% | 2.90x | 75.6% | 2.83x |
> |       | HASS | 96% | 2.61x | 86% | 2.05x | 75% | 2.05x | 88% | 1.96x | 74% | 1.94x | 83.8% | 2.12x |
> |       | HASS + TriSpec | 95% | **3.01x** | 84% | **2.51x** | 77% | **2.54x** | 89% | **2.31x** | 72% | **2.25x** | 83.4% | **2.52x** |
> |       | EAGLE3 | 98% | 2.78x | 84% | 2.55x | 77% | 2.52x | 86% | 2.58x | 74% | 2.52x | 83.8% | 2.59x |
> |       | EAGLE3 + TriSpec | 96% | **3.33x** | 86% | **3.06x** | 76% | **3.13x** | 88% | **2.94x** | 73% | **2.93x** | 83.8% | **3.08x** |
> |   DSQ    | Single Model (Target,32B) | 92% | 1.00x | 88% | 1.00x | 77% | 1.00x | 88% | 1.00x | 76% | 1.00x | 84.2% | 1.00x |
> |       | Single Model (Proxy,1.5B) | 71% | 2.99x | 78% | 2.91x | 66% | 3.15x | 62% | 3.12x | 47% | 3.11x | 64.8% | 3.06x |
> |       | HASS | 90% | 2.97x | 87% | 2.19x | 76% | 2.33x | 90% | 1.98x | 75% | 2.04x | 83.6% | 2.30x |
> |    | HASS + TriSpec | 89% | **3.27x** | 87% | **2.47x** | 77% | **2.74x** | 87% | **2.12x** | 73% | **2.18x** | 82.6% | **2.56x** |
> |       | EAGLE3 | 92% | 3.14x | 85% | 2.40x | 77% | 2.53x | 89% | 2.23x | 76% | 2.29x | 83.8% | 2.52x |
> |       | EAGLE3 + TriSpec | 88% | **3.40x** | 87% | **2.79x** | 79% | **2.92x** | 88% | **2.36x** | 72% | **2.42x** | 82.8% | **2.78x** |
> |   DSL    | Single Model (Target,70B) | 87% | 1.00x | 88% | 1.00x | 75% | 1.00x | 87% | 1.00x | 80% | 1.00x | 83.4% | 1.00x |
> |    | Single Model (Proxy,8B) | 72% | 3.80x | 81% | 3.92x | 70% | 3.95x | 79% | 3.93x | 60% | 3.96x | 72.4% | 3.91x |
> |       | EAGLE3 | 85% | 2.79x | 88% | 1.98x | 76% | 2.05x | 88% | 2.01x | 81% | 1.96x | 83.2% | 2.16x |
> |       | EAGLE3 + TriSpec | 86% | **3.93x** | 89% | **2.74x** | 78% | **3.06x** | 89% | **2.32x** | 78% | **2.39x** | 84.0% | **2.89x** |

---

### Author Response · Authors · 2025-11-21
**Summary of Our Responses & Paper Revisions**

We sincerely thank all the reviewers for their valuable and constructive reviews. We appreciate that the reviewers acknowledged TriSpec as a novel (Reviewer KuyU), simple and effective framework (Reviewers N6ke, iPrA), offering a new angle on speculative decoding (Reviewer EJNW), well-written and easy to follow (Reviewers KuyU, N6ke, EJNW), and recognized its strong motivation and effectiveness (Reviewers KuyU, N6ke, iPrA, EJNW).

1. **Experiments under different temperature settings:** In response to requests for performance evaluation across varied temperatures, we added experiments with temperature = 1 in Table 2. The results further confirm that TriSpec maintains both efficiency and accuracy under different temperature configurations.
2. **Cross–model family evaluation:** To examine architecture generality, we extended Tables 1–3 by adding results for the DeepSeek-R1-LLaMA family. Results show that TriSpec achieves the best accuracy–speed trade‑off under LLaMA architecture, demonstrating its architecture‑agnostic effectiveness.
3. **General and difficult task scenarios:** To address questions on broader and challenging settings, we added experiments on SpecBench, HotpotQA, and PolyMATH in Appendix B.1. Table 6 reports overall results, and Figure 5 shows SpecBench per‑category performance, demonstrating TriSpec is also effective across general and challenging task scenarios.
4. **Additional baseline methods:** We expanded the Related Work section to include reviewer‑mentioned approaches. In Table 1, we added two new deferral rules for SpecCascade as baselines, further demonstrating TriSpec’s superior performance.
5. **Full‑test‑set evaluation:** Responding to concerns about small evaluation sets and missing statistical measures, we provided explanations in Appendix B.2. Table 7 and Figure 6 now include full‑test‑set results and error bars from repeated‑subset evaluations, confirming the robustness of our findings.
6. **Proxy selection analysis:** To address questions about proxy choice, Appendix C.4 presents a detailed analysis. Figure 10 compares match rates between different‑size proxy models and the target model, validating the rationality of our chosen proxy configuration.

The major revised contents in the manuscript are highlighted in blue. Point-by-point responses to specific comments are given in the reviewer-specific sections below.

---

### Author Response · Authors · 2025-12-03
**Rebuttal Summary by the Authors**

Dear Area Chair,

We sincerely thank you for taking the time to handle our submission. To facilitate your assessment of our submission, we provide this concise summary to help you quickly understand the paper as well as the key points from our rebuttal and the corresponding revisions.

In this work, we propose **TriSpec**, a novel speculative decoding framework that adopts a new perspective that focuses on reducing the per-round verification cost and thereby achieves additional speedups within state-of-the-art speculative decoding methods such as EAGLE3. By introducing a lightweight proxy to partially replace the target model during verification, TriSpec reduces up to **50%** of target-model calls and achieves roughly **30%** additional speedup over standard speculative decoding, while remaining nearly lossless in accuracy. We conduct extensive experiments across diverse models and benchmarks, demonstrating that TriSpec is robust, scalable, and generalizes well.

Below, we summarize the key concerns raised by the reviewers and how we addressed them during the rebuttal phase.

1. **Clarification of novelty (Re: KuyU, iPrA):** We clarified the novelty of our work along three dimensions: experimental setting, optimization objective, and architecture. For each of these aspects, we compared our approach against prior methods. In particular, we provided a detailed analysis of the limitations that existing relaxed verification techniques encounter when combined with effective single-layer drafters, which directly motivates our proposed perspective and method.

2. **Robustness to temperature (Re: KuyU, iPrA):** We added experiments with sampling temperature = 1, showing that our method maintains both efficiency and accuracy under different sampling temperatures.

3. **Model families and scalability (Re: N6ke, iPrA, EJNW):** We extended our experiments to larger target models and different model families, including DeepSeek-R1-LLaMA-70B and LLaMA3.1-8B. The results indicate that TriSpec remains effective and efficient across model families and scales, directly addressing concerns about generalization and scalability.

4. **Broader and more challenging benchmarks (Re: N6ke, iPrA, EJNW)**: We expanded our evaluation to both broader and more difficult benchmarks, adding SpecBench and HotpotQA as general benchmarks, and AIME24, AIME25, GPQA-diamond, and PolyMath as more challenging tasks. These results demonstrate that our method is applicable and robust across diverse tasks and higher difficulty levels.

5. **The non-lossless issue (Re: EJNW):** We clarified that relaxing strict losslessness in speculative decoding to obtain further speedups has become a recognized research line, and we positioned TriSpec within this line of work. We also explained in detail that TriSpec allows tuning the margin threshold to achieve a more favorable speed–accuracy trade-off, and that in scenarios with stricter accuracy requirements it can revert to standard speculative decoding.

6. **Memory footprint and deployment (Re: iPrA, EJNW):** We quantified the small additional memory overhead introduced by the proxy and explained that TriSpec can be integrated into existing systems by closely following standard speculative decoding pipelines, thereby addressing concerns about practicality and deployment.

7. **Other points:** We also provided detailed responses to the remaining issues, including the impact of sampling temperature on the margin-based criterion (Re: KuyU), the choice and design of the proxy (Re: N6ke), and the evaluation metrics and training cost (Re: iPrA), among others.

Unfortunately, the discussion phase ended prematurely: we did not receive any follow-up from Reviewer **KuyU**, and although Reviewers **N6ke**, **iPrA**, and **EJNW** raised additional questions, our detailed responses could not be further discussed before the system freeze. Nevertheless, we believe that the added clarifications, further analyses, and supplementary experiments have substantially improved the manuscript and have comprehensively addressed the concerns raised by the reviewers. We respectfully ask you to take these concrete improvements into account in your final assessment.

Best regards,

The Authors

---

### Meta-Review · Area_Chair_BkQw · 2026-01-14

**Summary:**

Reviewers agreed that the paper is clearly written, well motivated, and that focusing on verification cost is a meaningful angle for speculative decoding. The central concern driving the recommendation is that TriSpec achieves its gains by giving up the lossless guarantee that makes speculative decoding a drop-in "last mile" accelerator, and multiple reviewers viewed this as a substantial limitation for practical adoption. A second theme was that the main idea appears close to existing hierarchical or multi-level speculative decoding and relaxed-verification lines, so the novelty bar depends heavily on whether the paper convincingly demonstrates a distinct optimization target, clear incompatibilities of prior methods with modern single-layer drafters, and compelling end-to-end evidence. Finally, reviewers raised practicality and evaluation concerns: whether the tri-model architecture can be deployed in mainstream inference stacks without substantial engineering overhead, whether memory and parallelism constraints under realistic throughput and long-context regimes undermine the gains, and whether the reported results are comparable and reproducible given questions about benchmark harnesses, truncation settings, and discrepancies with "official" baseline numbers. Overall, while the approach is promising and the speedups are nontrivial, the discussion converged on non-losslessness, deployability, and evaluation comparability as the key blockers to acceptance.

**Reviewer Concerns:**

The rebuttal and revision addressed several scope and validation issues. The authors expanded experiments across additional model families and sizes, including LLaMA-family targets and larger targets, added broader and more challenging benchmarks (including general and open-ended evaluations as well as harder reasoning suites), and reported full-test-set results and repeated-subset evaluations with variability estimates. They also added or discussed relevant baselines and clarified where prior relaxed-verification methods tend to be less compatible with very small single-layer drafters. At the same time, several core concerns remain partially unresolved. The novelty question is improved by the added discussion and baselines, but some reviewers still view the core mechanism as a straightforward extension of existing cascaded or verification-side designs. The non-lossless aspect remains a central limitation for some reviewers, because the method’s acceleration relies on proxy-approved tokens that can differ from target-only decoding under strict verification, which restricts drop-in applicability in settings that require output-distribution fidelity. There are also remaining questions about evaluation comparability and implementation details, including discrepancies with reported official benchmark numbers explained by truncation and settings choices, and the extent to which judge-based results and specific benchmark harnesses reflect standard practice. Finally, the deployment discussion remains partly aspirational: while the authors argue the architecture can be integrated with existing pipelines with modest effort and provide memory/runtime measurements, reviewers raised unresolved concerns about framework support, prefill–decode disaggregation, and engineering complexity under tensor parallelism and high-concurrency long-context workloads.

**Reviewer Scores:**

Reviewer KuyU’s main issues were positioning relative to hierarchical speculative decoding (notably TriForce) and robustness to different sampling temperatures. The revision adds explicit discussion of related hierarchical methods and provides new temperature experiments, which addresses these points. The remaining reservations are mostly about distinctness relative to adjacent work, so the most likely change is from 4 to 5, with a smaller possibility of remaining at 4 if novelty is still weighted heavily.

Reviewer N6ke’s concerns centered on generality beyond Qwen-based families, breadth beyond math/code, and the initial reliance on small subsets without uncertainty reporting. The rebuttal adds LLaMA-family experiments (including base LLaMA), expands to more general domains and harder tasks, and includes full-test-set evaluations and error bars. Given the reviewer’s follow-up indicating most concerns were addressed, a move upward from 6 is possible.

Reviewer iPrA raised concerns about limited novelty versus verification-side and multi-level SD work, the implications of dropping losslessness without strong stress tests in open-ended settings, missing related-work baselines, questions about evaluation methodology and discrepancies with official numbers, and practical deployment/memory issues for a three-model stack. The authors responded with additional baselines, broader evaluations (including judge-based evaluation), added harder benchmarks, and an explanation for metric discrepancies tied to truncation and settings, along with extra runs allowing longer outputs, plus memory/runtime measurements and a deployment sketch. However, this reviewer maintained substantive concerns after the follow-ups, so the most likely outcome is 4.

Reviewer EJNW’s concerns were generality beyond Qwen/DSQ, limited evidence on open-ended generation fidelity, the lossless guarantee as a key practical advantage of speculative decoding, and added deployment/memory complexity. The revision addresses generality and breadth with added LLaMA-family experiments and additional benchmarks and quantifies memory/runtime overhead, but the reviewer explicitly maintained a negative assessment emphasizing losslessness and deployment feasibility. The most likely outcome is that the score remains at 4, with a small chance of moving to 5 if the expanded experiments and overhead analysis are weighed more heavily than the lossless and deployment constraints.

---

### Decision · Program_Chairs · 2026-01-26

Reject